# Derivation and utility of schizophrenia polygenic risk associated multimodal MRI frontotemporal network

Shile Qi [1] ✉, Jing Sui [2] ✉, Godfrey Pearlson[3], Juan Bustillo[4], Nora I. Perrone-Bizzozero[4], Peter Kochunov[5], Jessica A. Turner[6], Zening Fu[7], Wei Shao[1], Rongtao Jiang[8], Xiao Yang[9], Jingyu Liu[7], Yuhui Du[10], Jiayu Chen[7] ✉, Daoqiang Zhang[1] ✉ & Vince D. Calhoun[7]

Schizophrenia is a highly heritable psychiatric disorder characterized by widespread functional and structural brain abnormalities. However, previous association studies between MRI and polygenic risk were mostly ROI-based single modality analyses, rather than identifying brain-based multimodal predictive biomarkers. Based on schizophrenia polygenic risk scores (PRS) from healthy white people within the UK Biobank dataset ($N = 22{,}459$), we discovered a robust PRS-associated brain pattern with smaller gray matter volume and decreased functional activation in frontotemporal cortex, which distinguished schizophrenia from controls with >83% accuracy, and predicted cognition and symptoms across 4 independent schizophrenia cohorts. Further multi-disease comparisons demonstrated that these identified frontotemporal alterations were most severe in schizophrenia and schizo-affective patients, milder in bipolar disorder, and indistinguishable from controls in autism, depression and attention-deficit hyperactivity disorder. These findings indicate the potential of the identified PRS-associated multimodal frontotemporal network to serve as a trans-diagnostic gene intermediated brain biomarker specific to schizophrenia.

Schizophrenia (SZ) is a severe psychiatric disorder with a lifetime risk of about 1%, that ranks among the leading causes of disability worldwide, with 26 million people affected[1]. The heritability of SZ is high, about 80–85%[2], estimated from twin and familial heritability studies[3], and ~24% with common single nucleotide polymorphism (SNP) heritability[4]. Genome wide association studies identified many SZ-related risk loci and SNPs that account for ~25% of the diagnostic variance[5,6], although the effect size of any single locus is small (~7%)[7]. Polygenic risk scores (PRS)[8] reflect cumulative risk by calculating from a weighted sum of common SZ genetic susceptibility loci, that represent an overall additive genetic vulnerability for developing schizophrenia, and provide a path to examine the underlying polygenic

[1]College of Computer Science and Technology, Nanjing University of Aeronautics and Astronautics, Nanjing, China. [2]State Key Laboratory of Cognitive Neuroscience and Learning, Beijing Normal University, Beijing, China. [3]Department of Psychiatry, Yale School of Medicine, New Haven, CT, USA. [4]Departments of Psychiatry and Behavioral Sciences, University of New Mexico, Albuquerque, NM, USA. [5]Department of Psychiatry, University of Maryland School of Medicine, Baltimore, MD, USA. [6]Department of Psychology, Georgia State University, Atlanta, GA, USA. [7]Tri-institutional Center for Translational Research in Neuroimaging and Data Science (TReNDS), [Georgia State University, Georgia Institute of Technology, Emory University], Atlanta, GA, USA. [8]Department of Radiology and Biomedical Imaging, Yale University, New Haven, CT, USA. [9]Huaxi Brain Research Center, West China Hospital of Sichuan University, Chengdu, China. [10]School of Computer & Information Technology, Shanxi University, Taiyuan, China. ✉e-mail: shile.qi@nuaa.edu.cn; jsui@bnu.edu.cn; jchen84@gsu.edu; dqzhang@nuaa.edu.cn

architecture of SZ and the impact of genetic factors on its neurobiological mechanisms[9].

In addition to genetic liability, SZ is also associated with widespread brain abnormalities[10–13], predominantly in frontotemporal, and thalamocortical areas[14,15]. Widely reported structural abnormalities include reduced thickness in frontotemporal cortices, and reduced hippocampal volumes[16]. Large, worldwide ENIGMA-schizophrenia case-control meta-analyses showed that SZ was associated with abnormalities in cortical thickness[17], subcortical volumes[18], and white matter integrity[15], that were highly replicable ($r = 0.8–0.95$) in independent cohorts[19,20]. These findings include: thinner cortical gray matter volume (GMV) and smaller cortical surface area, with the largest effect size in the frontal and temporal lobe[17], and decreased fractional anisotropy in anterior corona radiata, and corpus callosum[15], followed by smaller hippocampus, amygdala, and thalamus[18]. Additionally, cortical thinning in the superior temporal cortex is associated with positive symptoms[21], while prefrontal cortical thinning has been linked to negative symptoms in schizophrenia[22].

Patterns of associations between SZ PRS and MRI[23–28] have been reported for both structural[29] and functional[30] modalities, which presumably result in changes in psychological function and the clinical symptoms of SZ (none of which is exclusive to the disorder). Combining PRS and brain MRI phenotypes[31,32] may provide complementary insights into the underlying pathophysiological processes linked to schizophrenia from both genetic and brain phenotypic perspectives[6]. Higher polygenic burden was found to be linked with lower functional connectivity in the visual, default-mode, and frontoparietal networks based on task fMRI[33]. Higher SZ PRS were also associated with thinner frontotemporal cortices and smaller hippocampal subfield volume based on structural MRI (sMRI)[34]. Another recent study showed that PRS was correlated with reduced neurite density in 149 cortical regions, five subcortical structures, and 14 white matter tracts based on diffusion weighted MRI[29]. However, although brain abnormalities linked with PRS have been reported in schizophrenia, the aforementioned investigations focused on a single imaging modality and used a region of interest (ROI) based simple correlation analysis. There have not used, to the best of our knowledge, fusion of whole brain multimodal MRI to identify PRS-associated patterns, including the use of machine learning methods to assess its biomarker properties[35,36]. More specifically, there have been neither joint PRS-multimodal brain imaging studies focused on the classification of SZ and healthy controls (HC), nor the use of these variables to predict cognition or symptoms. If successful, the approach proposed here will be an important step towards the use of imaging-genetic data as potential biomarkers in assisting clinicians in differential diagnosis and in prediction of relevant clinical outcomes.

In this study, we hypothesized that PRS-SZ would be associated with a specific multimodal covarying brain pattern, and that these multimodal brain features would accurately distinguish SZ from healthy controls (HC), as well as predict major clinical measures for SZ. The UK Biobank (UKB, https://www.ukbiobank.ac.uk/) data were used as a discovery cohort to identify the PRS-associated multimodal brain pattern in healthy white people ($N = 22,459$) with resting state fMRI and sMRI. This set of potential biomarkers was then validated for diagnostic classification and cognitive/symptomatic prediction across four independent SZ cohorts. Our aims included: (1) identifying SZ PRS-associated fALFF + GMV (fractional amplitude of low frequency fluctuations) patterns in the large UKB population (Fig. 1a, b); (2) PRS-pattern validation within UKB using different population and $P_{SNP}$ thresholds (Fig. 1c); (3) classification and prediction abilities' validation of the PRS pattern in four independent SZ cohorts (Fig. 1d, e); and (4) SZ-specificity validation of the PRS pattern with respect to other psychiatric disorders (Fig. 1f). By combining PRS and the multimodal MRI features from the selected large UKB sample[39] and four independent SZ cohorts, we sought to identify a more robust PRS-associated functional-structural covarying MRI signature for SZ.

## Results

### SZ PRS-associated multimodal brain network

Schizophrenia PRS were calculated based on Psychiatric Genomics Consortium Schizophrenia (PGC SZ 2) 108 risk loci[7], thresholded at $P_{SNP} < 5.0e−08$ and pruned at $r^2 < 0.1$, which followed a normal distribution (Supplementary Fig. 1). Head motion, site, gender and age were regressed out from fALFF/GMV feature matrices prior to fusion analysis. Within the healthy white people UKB data ($N = 22,459$, demographic information can be found in Table 1), fusion with PRS was performed to identify PRS-associated fALFF + GMV multimodal pattern (details on fusion with reference method can be found in "Methods"). One joint component (Fig. 2a) was positively correlated with PRS ($r = 0.074$, $p = 4.1e−30^*$ for fALFF; $r = 0.074$, $p = 1.6e−28^*$ for GMV, Fig. 2b). * means false discovery rate (FDR) corrected for multiple comparisons, which represents the same meaning throughout the paper. Although the variance explained was <1% (consistent with previous SZ PRS and ROI-based single modality analysis for UKB[29,40]), the statistical power was high enough ($1 − β = 1$, Supplementary Fig. 2 and "Power analysis"). The direct correlation between SZ-PRS and voxel wise MRI features throughout the brain (60758 and 90638 voxels for fALFF and GMV) was calculated. The maximum absolute correlation $r$ is only 0.03 and 0.028, and the mean $r$ is 0.008 and 0.0006 for fALFF and GMV respectively. Apart from the voxel wise correlation between SZ-PRS and MRI features, we also tested the correlation between the mean values extracted from ALL atlas and SZ-PRS for both fALFF and GMV under different $P_{SNP}$ thresholds (5.0e−08, 1.0e−04, 0.05). Results (Supplementary Figs. 3–5 and Supplementary Table 1) showed that the variance explained was <1% for all the brain areas under 3 different $P_{SNP}$ thresholds.

The red/blue brain regions indicate positive/negative correlation with PRS in fALFF or GMV, i.e., red fALFF/GMV in the identified brain areas positively correlate with PRS. PRS-associated pattern includes positive fALFF in middle and inferior frontal cortex (MIFC), superior and middle temporal cortex (SMTC), negative fALFF in thalamus, posterior cingulate cortex (PCC), middle occipital cortex (MOC), and lingual gyrus (LG), accompanied with positive GMV in anterior insula and hippocampus, and negative GMV in middle insula, superior/middle/inferior temporal cortex (SMITC), fusiform gyrus (FG) and parahippocampus. The identified brain regions are summarized in Supplementary Table 2 for fALFF and GMV in Talairach labels, respectively.

### PRS-MRI linkage replication in SZ patients

By linear projecting the identified PRS spatial maps onto SZ patients ($N = 290$), the correlation between PRS and the component remain significant ($r = 0.35$, $p = 1.2e−04^*$ for fALFF; $r = 0.33$, $p = 1.4e−04^*$ for GMV, Fig. 2c), which means that the association between PRS and the PRS-pattern can be replicated in an independent SZ dataset (details are in Supplementary "Linear projection"). In order to confirm that the extracted PRS-associated pattern is specific to the PRS measure but not a random pattern, we permuted the PRS in the supervised fusion analysis (details are in Supplementary "Null pattern"). Note that the random pattern (Supplementary Fig. 6b) was very dissimilar to the identified PRS-associated pattern, confirming its specificity to the PRS, but not a null pattern.

### PRS-pattern consistency across PRS parameters within UKB sample

The robustness of the identified PRS pattern was also validated. The same PRS-guided fusion was performed on different split of UKB sample (healthy white people, healthy subjects and all available subjects that passed MRI quality control) under different $P_{SNP}$ (5.0e−08, 1.0e−04, 0.05) and pruning ($r^2 < 0.1$ and 0.2) thresholds. Results showed that the identified PRS-associated frontotemporal pattern was highly replicable within UKB (Fig. 3) under different $P_{SNP}$ (Supplementary Fig. 7). The positive fALFF in MIFC, SMTC, negative fALFF in

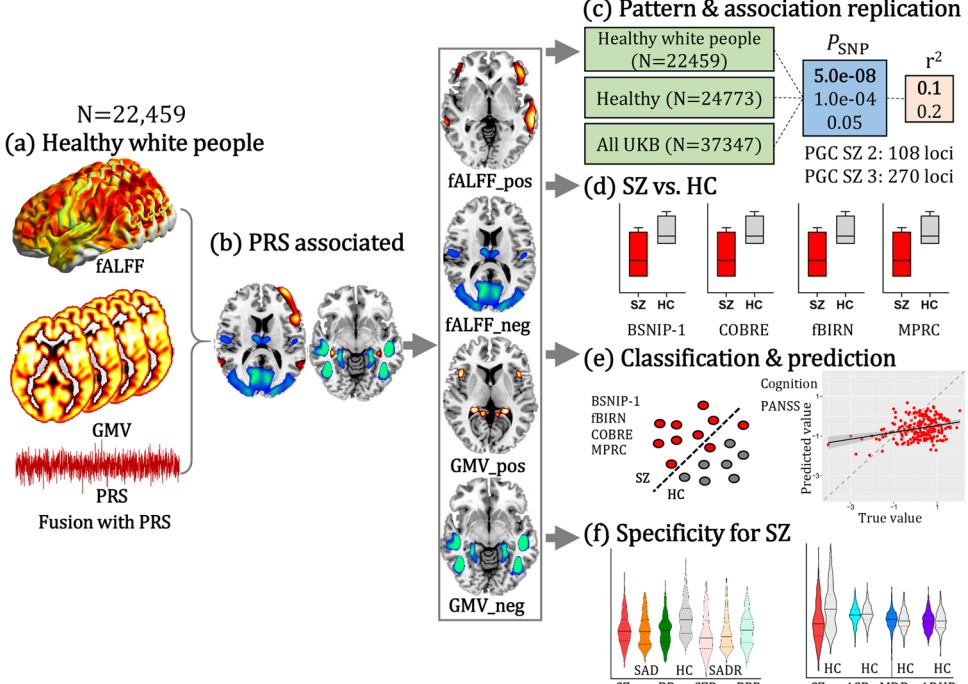

**Fig. 1 | Flowchart for developing, testing and validating the SZ PRS-associated multimodal pattern. a** SZ PRS was used as a reference to guide fALFF + GMV fusion to identify PRS-associated multimodal networks. **b** PRS-associated networks were separated as positive (Z > 0) and negative (Z < 0) brain regions based on the Z-scored brain maps, resulting in 4 features for the following analysis. **c** The same fusion with PRS analysis was performed on healthy white people, healthy subjects, and all the available subjects within UKB with PRS thresholded at $P_{SNP}$ = 5.0e−08, 1.0e−04 and 0.05, and pruned at $r^2$ < 0.1 and 0.2, respectively. **d** Group differences between SZ and HC of the identified PRS-associated features were tested across 4 independent SZ cohorts (including BSNIP-1, COBRE, fBIRN and MPRC). **e** Linear SVM was used to classify SZ and HC across 4 independent SZ cohorts based on PRS-associated features (including BSNIP-1, COBRE, fBIRN and MPRC); multiple linear regression was performed to predict cognition and symptom for SZ across 3 independent cohorts (BSNIP-1, COBRE, and fBIRN). MPRC data were not included in the prediction analysis since the related clinical measures were not available. **f** Group differences between SZ and other psychosis subjects and their relatives, and between other psychiatric disorders (ASD, MDD, ADHD) and HC were tested. UKB (UK Biobank, N = 37,347); BSNIP-1 (Bipolar and Schizophrenia Network for Intermediate Phenotypes), SZ: N = 178; HC: N = 220; COBRE (Center for Biomedical Research Excellence), SZ: N = 100; HC: N = 90; fBIRN (Functional Imaging Biomedical Informatics Research Network), SZ: N = 164; HC: N = 157; MPRC (Maryland Psychiatric Research Center), SZ: N = 164; HC: N = 157; ASD autism spectrum disorder, MDD major depressive disorder, ADHD attention-deficit/hyperactivity disorder, SVM support vector machine.

thalamus, PCC, MOC, and LG, accompanied with positive GMV in anterior insula and hippocampus, and negative GMV in middle insula, SMITC, FG and para-hippocampus were replicated (>80%, spatial similarity, Supplementary Tables 3, 4) with different pruning thresholds under different population selection strategies. The positive fALFF in MIFC, SMTC, negative fALFF in PCC and MOC, accompanied with positive GMV in anterior insula and hippocampus, and negative GMV in MITC, and para-hippocampus were validated (>50%) under different $P_{SNP}$ thresholds (Supplementary Fig. 7). Details on sample selection and the spatial similarity calculation can be found in Supplementary Fig. 8 and Supplementary "Spatial similarity". Furthermore, we also calculated the PRS based on the preprint PGC SZ 3 (270 loci, identified by the largest extant GWAS dataset with N = 69,369 SZs and N = 236,642 HCs)[4]. Results showed that SZ PRS were highly correlated between PGC3 and PGC2 (Supplementary Fig. 9), and the PRS-associated pattern was similar (0.89 and 0.85 for fALFF and GMV components, Supplementary Fig. 10) between PGC2-guided and PGC3-guided fusion.

### Site and motion effects on the identified PRS-pattern

For the MRI imaging data, there are three sites available in UKB, including Cheadle, Reading and Newcastle. We performed the same PRS-guided fusion for each site separately to test the similarity of the identified PRS-associated frontotemporal pattern. Results (Supplementary Fig. 11 and Supplementary Tables 5, 6) showed that there was high percentage of spatial similarity among Cheadle, Reading, Newcastle and UKB (>0.70). Note that site was regressed out from fALFF/

GMV feature matrices prior to the primary fusion analysis. Thus we do not believe that site would be a major confounding factor with respect to the identified PRS frontotemporal multimodal pattern.

To control confounding effects of motion artifact, several strategies were conducted. In the preprocessing procedure for fMRI, we despiked the fMRI data: nuisance covariates (6 head motions + cerebrospinal fluid [CSF] + white matter [WM] + global signal) were regressed out. Outlier subjects with framewise displacements (FD) exceeding 1.0 mm, and head motion exceeding 2.5 mm of maximal translation (in any direction of x, y, or z) or 1.0° of maximal rotation were excluded in the primary analysis. Furthermore, we also excluded subjects with >0.2 mm FD to get a subset of UKB (N = 13,490, 60% subjects' head motion <0.2 mm) to perform PRS-guided fusion analysis to test whether the identified multimodal frontotemporal pattern was impacted by head motion. Result showed that the identified PRS-associated pattern (frontotemporal cortex and thalamus in fALFF, accompanied with thalamus, hippocampus, para-hippocampus and temporal cortex in GMV) can be replicated on UKB subset with FD < 0.2 mm (Supplementary Fig. 12b). Considering there is no group difference in head motion between SZ and HC (p > 0.1, Supplementary "Group differences of mean FD between SZ and HC"), and no significant correlation between mean FD and PRS (Supplementary Table 7), and the partial correlation between the identified component and PRS still significant after regressing out mean FD (Supplementary "Partial correlation"), and the PRS-pattern can be replicated on UKB subset with head motion <0.2 mm, the current fusion analysis was based on fALFF not functional connectivity (which is sensitive to head

**Table 1 | Demographic information of subjects participated in this study**

| Datasets | Group | Number | Age (mean/sd) | Gender (M/F) |
|---|---|---|---|---|
| UKB | Healthy white people | $N = 22,459$ | 64.3/7.5 | 11,173/11,286 |
| BSNIP-1 | SZ | $N = 178$ | 34.5/12.0 | 124/54 |
| | SAD | $N = 134$ | 36.3/12.3 | 58/76 |
| | BP | $N = 143$ | 36.2/13.2 | 47/96 |
| | HC | $N = 220$ | 38.8/12.6 | 90/130 |
| | SZR | $N = 162$ | 42.6/15.3 | 57/105 |
| | SADR | $N = 149$ | 39.9/16.0 | 52/97 |
| | BPR | $N = 142$ | 40.4/15.9 | 52/90 |
| COBRE | SZ | $N = 100$ | 38.5/14.1 | 78/22 |
| | HC | $N = 90$ | 38.0/11.6 | 65/25 |
| fBIRN | SZ | $N = 164$ | 39.1/11.4 | 122/42 |
| | HC | $N = 157$ | 37.5/11.3 | 112/45 |
| MPRC | SZ | $N = 224$ | 37.9/13.8 | 87/137 |
| | HC | $N = 137$ | 41.1/15.6 | 80/57 |
| ABIDE II | ASD | $N = 421$ | 13.4/5.6 | 421/0 |
| | HC | $N = 389$ | 13.7/6.2 | 389/0 |
| Depression | MDD | $N = 260$ | 32.8/11.0 | 99/161 |
| | HC | $N = 281$ | 31.3/10.6 | 102/179 |
| ADHD-200 | ADHD | $N = 346$ | 11.5/2.9 | 270/76 |
| | HC | $N = 478$ | 12.0/3.2 | 264/214 |

*UKB* UK Biobank, *BSNIP-1* Bipolar and Schizophrenia Network for Intermediate Phenotypes, *fBIRN* Functional Imaging Biomedical Informatics Research Network, *COBRE* Center for Biomedical Research Excellence, *MPRC* Maryland Psychiatric Research Center, *ABIDE* Autism Brain Imaging Data Exchange, *M/F* male/female, *SAD* schizophrenia affective disorder, *BP* psychotic bipolar disorder, *SZR* schizophrenia relatives, *SADR* schizophrenia affective disorder relatives, *BPR* bipolar disorder relatives, *ASD* autism spectrum disorder, *MDD* major depressive disorder, *ADHD* attention-deficit/hyperactivity disorder.

motion[41–45]), mean FD was regressed out from fALFF/GMV feature matrices prior to fusion analysis, we believe that micro-motion is not a major factor affecting the current results. As for IQ, the direct correlation between PRS and IQ is marginally significant ($p = e^{-05}$, not FDR corrected), and the PRS-pattern after regressing out IQ is almost the same as the original PRS pattern (Supplementary Fig. 12d).

**Group differences of PRS-pattern between SZ and HC**
The identified PRS-associated fALFF + GMV components were separated into positive ($Z > 0$) and negative ($Z < 0$) brain networks based on the Z-scored brain maps. Thus 4 PRS-associated brain features (fALFF_positive, fALFF_negative, GMV_positive, GMV_negative) were obtained by averaging fALFF/GMV in these networks. Two-sample t-tests were used to estimate the group differences of these 4 PRS features between SZ and HC. Results (Fig. 4a) showed that fALFF values in both fMRI positive and negative networks, as well GMV in sMRI positive and negative networks were consistently significantly lower in SZ than in HC across BSNIP-1 (Bipolar and Schizophrenia Network for Intermediate Phenotypes), COBRE (Center for Biomedical Research Excellence), fBIRN (Functional Imaging Biomedical Informatics Research Network) and MPRC (Maryland Psychiatric Research Center) cohorts (details on the demographic information of these 4 SZ cohorts are in Table 1).

**Classification between SZ and HC**
The classification ability of the identified PRS-associated brain network was validated by using a linear support vector machine (SVM) approach to classify SZ patients and HCs. In addition to the averaged fALFF/GMV values, the first 5 principal components (PC, obtained from principal component analysis) resulted from decomposing the fALFF/

GMV positive/negative feature matrices within the identified PRS networks were also included as feature input, resulting in 6 features for each (the mean + 5 PC) PRS-associated network, i.e., 24 features in each HC-SZ cohorts (details on feature selection and SVM classification can be found in Methods and Supplementary "Feature selection and classification" sections). Note that the first 5 PCs captured 99% variance from the identified PRS-associated ROIs, while the mean only represented <50% variance (Supplementary Fig. 13 and Supplementary Table 8). Results (Fig. 4b) showed that the identified PRS-associated features can consistently classify between SZ and HC with >83% accuracy and >0.9 AUC in 4 independent SZ cohorts (BSNIP-1: ACC = 85.2%, AUC = 0.95; COBRE: ACC = 83.7%, AUC = 0.90; fBIRN: ACC = 89.9%, AUC = 0.96; MPRC: ACC = 84.4%, AUC = 0.94). The first PC contributed the most and followed by the mean (Supplementary Fig. 14), demonstrating the necessarily of adding the additional 5 PCs in classification. Note that the 5 PCs were extracted from the identified PRS-associated ROIs, not from the whole brain. While for the null pattern (Supplementary Fig. 6b), there were neither group difference between SZ and HC (Supplementary Fig. 15a), nor classification ability to discriminating SZ and SZ (Supplementary Fig. 15b). These results demonstrated the specificity of the identified PRS-pattern in classifying between SZ and HC. The classification accuracies were approximated as around 50% as a random distributed accuracy when treating sites as categories (the more number of sites the lower classification accuracy, Supplementary Fig. 16), indicating that site was not a major confounding factor for the current SZ-HC classification analysis.

**Prediction of cognition and symptom for SZ**
The four mean PRS-associated brain features were further used to construct multiple linear regression models (Eq. 2) for each domain from COBRE cohort to predict cognitive and symptom measures for fBIRN and BSNIP SZ patients (the same models and the same ROIs were used in the generalized prediction). Correlations between the estimated symptom/cognitive scores and its true values were calculated to estimate the prediction performance. These four PRS features predicted attention, working memory and composite cognition, as well as PANSS negative scores for all three independent SZ cohorts (Fig. 5). Pearson correlations of $r = 0.54, 0.52, 0.53, 0.48$ were achieved between the estimated composite/attention/memory/PANSS_N and its true values for COBRE; $r = 0.44, 0.49, 0.52, 0.49$ for BSNIP-1 cohort; $r = 0.51, 0.44, 0.55, 0.51$ for fBIRN. MPRC was not included in the prediction since symptom and cognitive data were not available for this cohort. Note that the predicted cognition was measured using 3 different cognitive batteries (Supplementary Tables 10–12, BSNIP-1: Brief Assessment of Cognition in Schizophrenia, BACS; fBIRN: Computerized Multiphasic Interactive Neuro-cognitive System, CMINDS; COBRE: Measurement and Treatment Research to Improve Cognition in Schizophrenia Consensus Cognitive Battery, MCCB)[46].

**Specificity of PRS pattern among psychosis and their relatives**
To test whether the identified PRS-derived pattern was evident in other psychotic disorders, two-sample t-tests were used to calculate group differences of the 4 PRS features within the BSNIP-1 cohort among SZ ($N = 178$), schizo-affective disorder (SAD, $N = 134$), psychotic bipolar disorder (BP, $N = 143$), HC ($N = 220$), schizophrenia relatives (SZR, $N = 162$, first degree relatives), schizo-affective disorder relatives (SADR, $N = 149$), and bipolar disorder relatives (BPR, $N = 142$), after correcting for site effects. Results showed that fALFF in positive and negative networks were lower in psychotic disorders and their relatives than HC (Fig. 6). While for sMRI features, GMV in both positive and negative networks were significantly decreased in SZ and SADR than HC.

**Specificity of PRS pattern among SZ, ASD, MDD, and ADHD**
The ability of the identified PRS-associated brain features in differentiating between other neuropsychiatric/mood disorders and HC was

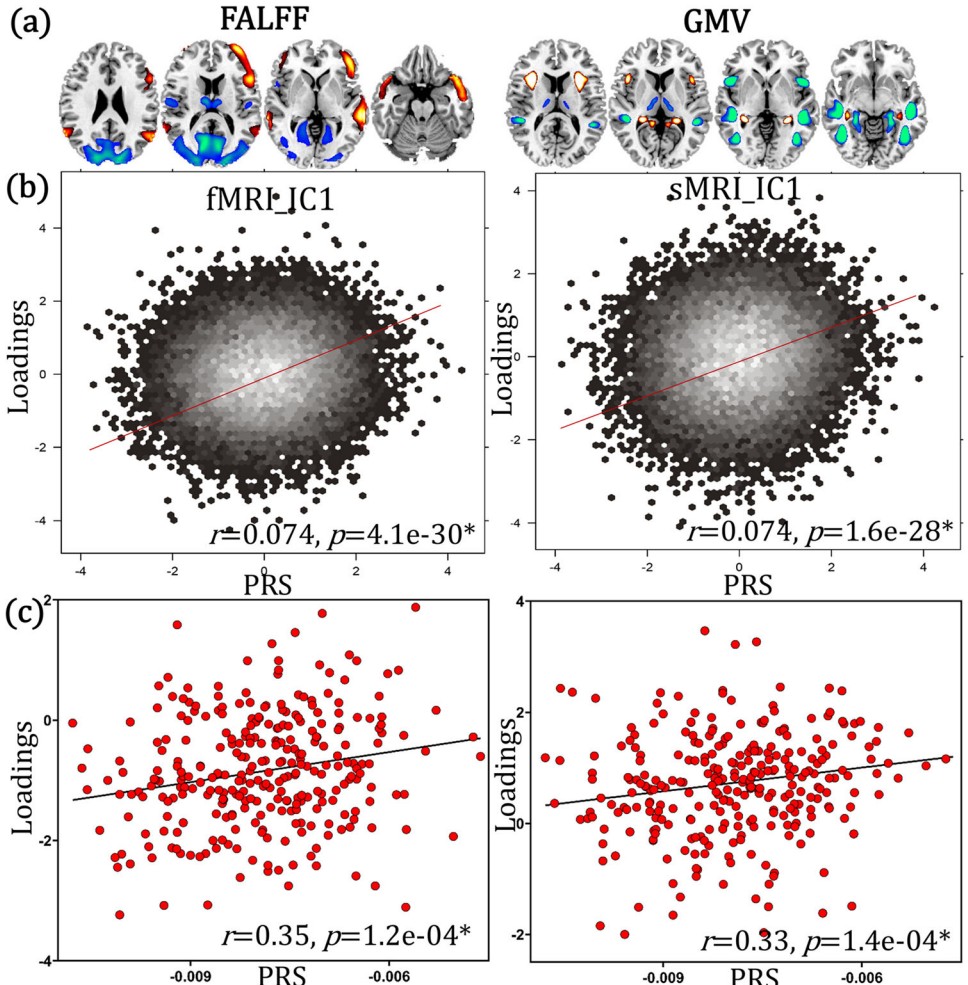

**Fig. 2 | The identified PRS-associated multimodal joint components in UKB healthy white people (***N*** = 22,459). a** Spatial brain maps visualized at |Z| > 2. **b** 2D density plot of PRS and loadings of components in UKB (*p* = 4.1e−30* and *p* = 1.6e−28* for fMRI and sMRI respectively). **c** Correlations (*p* = 1.2e−04* and *p* = 1.4e−04* for fMRI and sMRI respectively) between projected (projecting spatial maps from UKB to SZ patients to obtain an estimation of their mixing matrices) loadings and PRS within SZ patients (*N* = 290, where PRS was available). Pearson correlation analysis was used to calculated the correlation between PRS and loadings in (**a**, **b**). Source data are provided as a Source Data file.

also tested by two-sample *t*-tests. Results (Fig. 7) showed that PRS-associated frontotemporal pattern for individuals with autism spectrum disorder (ASD, *N* = 421), major depressive disorder (MDD, *N* = 260) and attention-deficit/hyperactivity disorder (ADHD, *N* = 346) did not differ significantly from the HC group within each cohort, but were only significantly reduced in SZ. The direct comparison among SZ, ASD, MDD and ADHD (ANOVA) can be found in Supplementary Fig. 17.

## Discussion

In this study, we developed a PRS-associated multimodal imaging pattern, validated its utility for classification and prediction, and elucidated frontotemporal abnormalities in psychosis, indicating that PRS-associated pattern may be a potential biomarker of genetic risk associated with brain differences that contributes to the pathogenesis of schizophrenia and other psychoses. Specifically, by combining genetic risk and multimodal MRI data (fALFF + GMV) in the large discovery UKB dataset (healthy white people, *N* = 22,459), we identified a PRS-associated multimodal frontotemporal pattern. This PRS pattern was highly replicable within UKB under different PRS $P_{SNP}$ and pruning thresholds with different population selection criteria (>80% similarity) for both 108 loci and 270 loci. The identified PRS frontotemporal network can further differentiate SZ from HC across four independent SZ cohorts with an accuracy of >83%, and even predict cognition and

negative symptom severity successfully across three SZ cohorts where such data were available. Furthermore, we demonstrated that the identified PRS-pattern had fairly high specificity to schizophrenia, moderate sensitivity to psychosis, but was not sensitive to the diagnoses of ASD, MDD or ADHD. To the best of our knowledge, this is the first study to further evaluate the biomarker property of the PRS-associated multimodal brain networks through rigorous cross-site classification and prediction, which may inform more about genetic risk that SZ associated confounding.

A major finding of the current study was the identification of a PRS-associated multimodal frontotemporal network, which involved: decreased fALFF in MIFC, SMTC, thalamus, PCC, MOC, and LG, covarying with reduced GMV in insula, SMITC, FG and the hippocampal complex in SZ. This network was robustly decreased across four independent SZ cohorts. These findings are consistent with the ENIGMA-schizophrenia meta-analyses which reported the largest effect sizes for frontal and temporal lobe reductions relative to other cortical regions[17] and for hippocampus relative to other subcortical areas[18]. PRS and functional connectivity studies showed that higher PRS was correlated with lower functional connectivity in frontoparietal[33] and frontotemporal[47] systems. Another PRS and structural MRI association study showed that PRS was correlated with structural MRI phenotypes in temporal, cingulate, and prefrontal

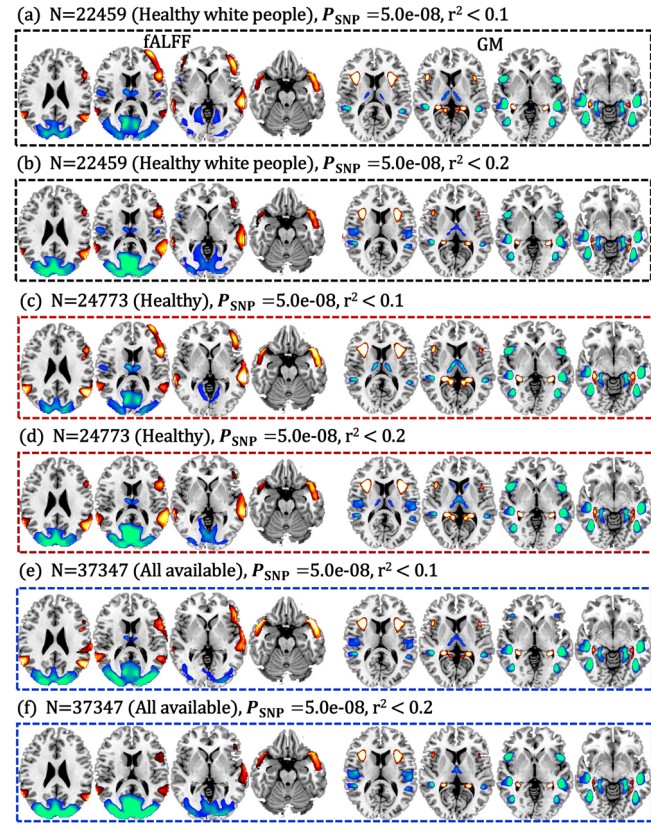

(a) N=22459 (Healthy white people), $P_{SNP}$ =5.0e-08, r² < 0.1

(b) N=22459 (Healthy white people), $P_{SNP}$ =5.0e-08, r² < 0.2

(c) N=24773 (Healthy), $P_{SNP}$ =5.0e-08, r² < 0.1

(d) N=24773 (Healthy), $P_{SNP}$ =5.0e-08, r² < 0.2

(e) N=37347 (All available), $P_{SNP}$ =5.0e-08, r² < 0.1

(f) N=37347 (All available), $P_{SNP}$ =5.0e-08, r² < 0.2

**Fig. 3 | Replication of the identified PRS-associated pattern within UKB.** Fusion with PRS pruned at $r^2 < 0.1$ and pruned at $r^2 < 0.2$ on healthy white people ($N = 22,459$, **a**, **b**), on healthy participants ($N = 24,773$, **c**, **d**), and on all the available subjects in UKB that passed MRI quality control ($N = 37,347$, **e**, **f**). There is high spatial overlap (>80%) among these PRS-associated patterns (details can be found in the Supplementary "Spatial similarity" section). PRS-pattern comparisons (>50%) among $P_{SNP} < 5.0e{-}08$, $1.0e{-}04$, $0.05$ can be found in Supplementary Fig. 7.

cortex, insula, and hippocampus[29]. In addition, higher PRS was found to be associated with thinner frontotemporal cortices and smaller hippocampal subfield volume[34]. All these previous single modality studies are consistent with our current identified PRS-associated multimodal pattern in a frontotemporal network and subcortical areas (including hippocampal complex, thalamus and insula) connecting them. More importantly, the identified PRS-associated frontotemporal pattern was robustly replicable with >80% spatial similarity within UKB under different conditions (different PRS $P_{SNP}$ and pruning thresholds, different population selection, different PGC versions), providing converging and complementary supporting evidence of genetic variants on both brain function and structure in SZ. Despite reliable PRS-pattern validation, the PRS was only weakly associated with the components' loadings, but with high enough statistical power ($1 - \beta = 1$). This is consistent with most previous published large sample sized UKB SZ PRS-MRI association studies[29,40], as well as in Nature 2022 reported that smaller (sample size) brain wide association studies have reported larger correlations than the largest effects measured in larger samples[48].

Another important contribution of the current investigation was the biomarker validation of this PRS-associated frontotemporal network in both diagnostic classification and symptom/cognition prediction, which previous PRS and MRI phenotype association studies did not pursue. We hope that this work helps re-focus SZ researchers on the utility of genetic risk mediated brain multimodal biomarker identification. The ability of SZ PRS-associated multimodal frontotemporal related imaging features to differentiate between SZ and HC with >83%

accuracy across four independent SZ cohorts is important and potentially clinically significant. Although genetic data alone may have relatively small variance explanation for SZ[4,40], our results demonstrated that the PRS-associated multimodal MRI phenotypes can classify SZ from HC with AUC > 0.9 across varied samples. The predictive power of this PRS pattern was further validated by its ability to predict cognitive composite, attention and working memory (despite the use of three different cognition assessment batteries), as well as PANSS negative scores across three independent SZ cohorts. This is in line with that the PRS for schizophrenia are associated with negative symptoms and working memory performance[49]. The current study further indicates that SZ genetic liability associated frontotemporal abnormalities from structural and functional perspectives[27], may result ultimately in cognitive dysfunction (attention and memory) and altered clinical symptoms. We emphasize that it is the combination of genetic risk variants and its related intermediate MRI phenotype together that are effective in classification and prediction. Rigorous generalized classification and prediction validation across several new clinical cohorts with SZ is necessary before specific recommendations for clinical implementation can be made[50].

The specificity of the identified PRS-associated pattern was also validated among psychosis patients and their relatives, as well as individuals with ASD, MDD and ADHD. Particularly, SZ PRS-pattern characterized diagnostic heterogeneity within psychosis, where multimodal covarying frontotemporal alterations were most severe in SZ and SAD, milder in psychotic BP, and indistinguishable from HC in ASD, MDD and ADHD, demonstrating that the PRS-associated multimodal covarying frontotemporal pattern revealed consistent abnormalities across psychotic disorders (including SZ, SAD and BP with psychosis), but not for ASD, MDD and ADHD. This suggests that the SZ PRS-associated frontotemporal pattern is highly sensitive to schizophrenia, moderately sensitive to psychosis in general, and insensitive to other psychiatric disorders, consistent with that the PRS for schizophrenia are associated with liability for BP and SAD[51]. This underscores the ability of PRS pattern to transect classic diagnostic categories and accords with evidence of overlapping psychopathology between schizophrenia, SAD and BP. These results point to the importance and specificity of the identified PRS-multimodal frontotemporal network and subcortical regions connecting them, that underlie the common pathophysiological process among psychosis, including SZ, SAD and psychotic BP.

Some limitations should be considered. First, the UKB is an aging cohort of largely European descent that is on average wealthier and healthier than the general population[52]. However, we investigated the generalizability of the identified PRS-associated frontotemporal pattern in both classification and prediction on demographically diverse and independent SZ samples (including four cohorts: BSNIP-1, fBIRN, COBRE and MPRC). Second, all the datasets included in this study were collected from multiple sites[53]. However, site and other confounding factors including age, gender and head motion were regressed out from fALFF and GMV matrices prior to fusion analysis, which helped diminish possible confounds of site, age, gender and motion as major contributors for our current investigation (Supplementary "Motion effect" and "Site effect" sections). Finally, although the current study used static brain metrics (fALFF), dynamic functional connectivity that captured the temporal properties[54,55] of the brain signaling could also be investigated in a future study.

Collectively, the present study combined PRS, fMRI and sMRI to identify neuroimaging patterns associated with PRS[56] by supervised fusion, and tested the role of transdiagnostic properties by both classification and prediction, as well as its specificity to psychosis. A specific and robustly validated brain network involving frontotemporal, thalamus, insula and hippocampal complex appeared to underlie the genetic mechanisms impacting on multiple brain MRI phenotypes in SZ. Reduced fALFF and GMV in this network correctly

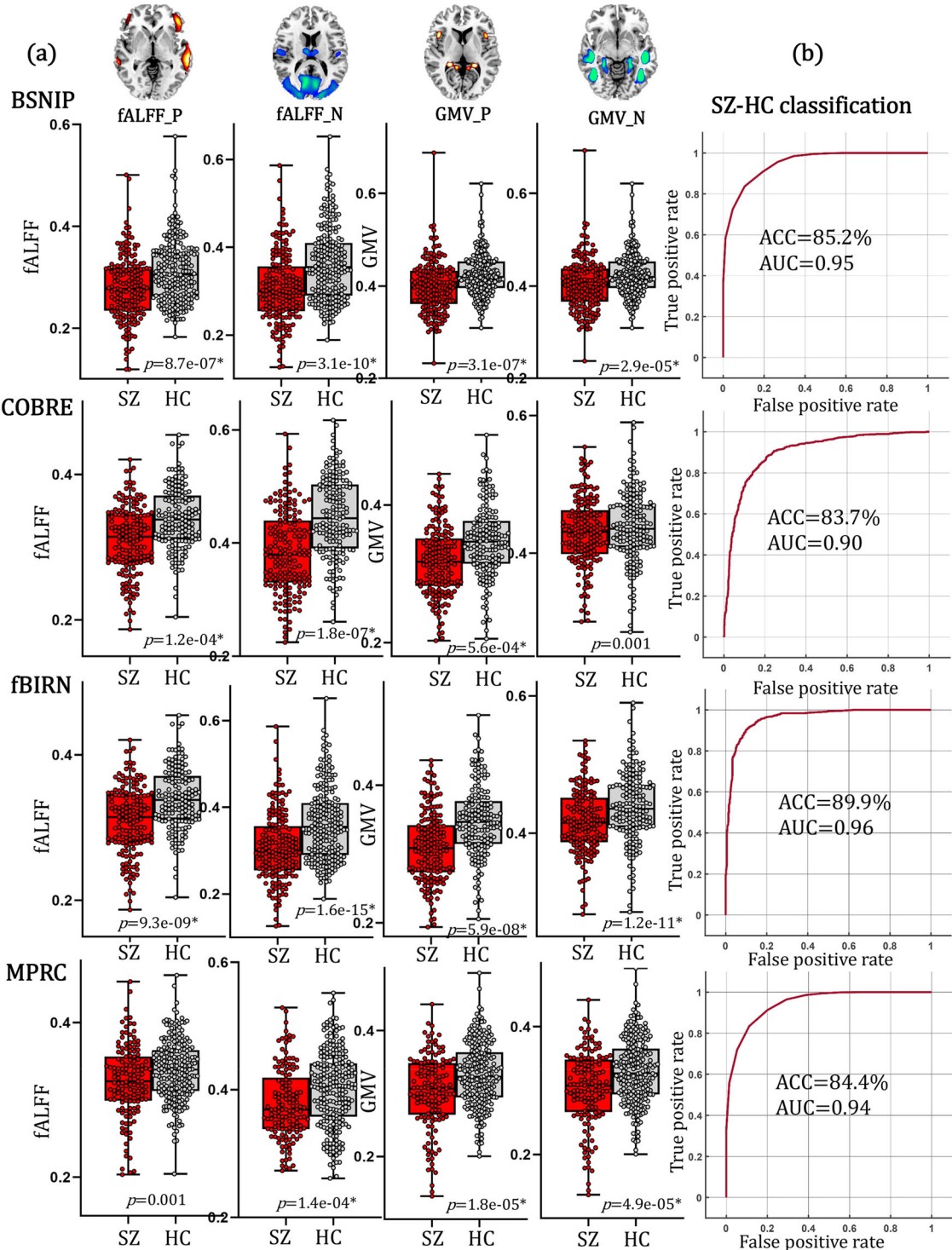

**Fig. 4 | The diagnostic ability of the identified PRS-associated frontal-temporal network. a** Group differences between SZ and HC of PRS-associated the positive and negative networks for independent BSNIP-1 (SZ: $N = 178$; HC: $N = 220$), COBRE (SZ: $N = 100$; HC: $N = 90$), fBIRN (SZ: $N = 164$; HC: $N = 157$) and MPRC (SZ: $N = 164$; HC: $N = 157$) cohorts, respectively. Two-tailed two-sample T test was used to calculate the group differences in (**a**). The minima, maxima and the mean were displayed in the box plots. **b** ROC curves of the classification results between SZ and HC for BSNIP-1, COBRE, fBIRN and MPRC cohorts, respectively. The classification accuracies were approximated as around 50% as a random distributed accuracy when treating site as categories (the more number of sites the lower classification accuracy, Supplementary Fig. 16), indicating that site was not a major confounding factor for the current SZ-HC classification analysis. AUC area under the curve, ACC accuracy.

classified SZ and HC, and also predicted cognition and negative symptoms in several independent SZ cohorts, highlighting the potential of PRS-associated multimodal neuroimaging pattern in biomarker development. Furthermore, the extant literature in the field of PRS and neuroimaging association studies in SZ has mainly focused on pathophysiology explanations of SZ, while our current study goes beyond PRS and neuroimaging associations by testing its ability in classification and prediction, as well as demonstrate its diagnostic

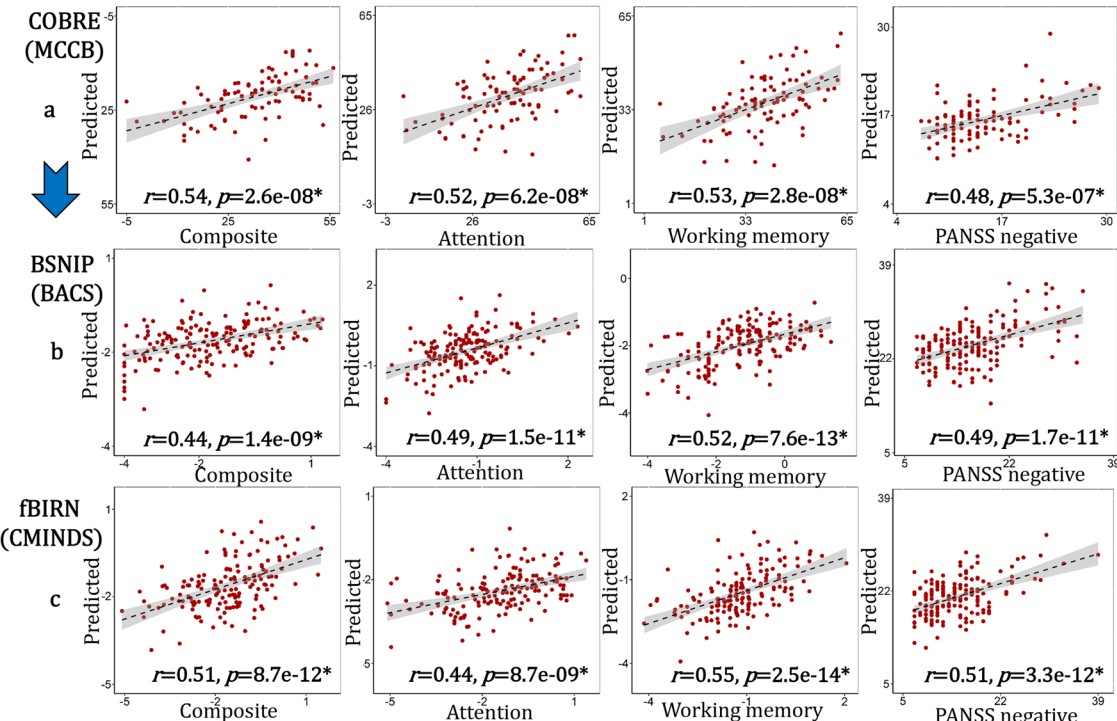

**Fig. 5 | Prediction results for cognition and symptom across BSNIP-1, fBIRN and COBRE based on the identified PRS-associated features.** The prediction models for each domain were trained based on single site cohort: COBRE. Then the same models and the same ROIs were used to predict cognition/symptom for BSNIP and fBIRN cohorts. Pearson correlation was used to calculated the correlation between the true values and the predicted values in (**a**–**c**). The gray regions in (**a**–**c**) indicate a 95% confidence interval. Source data are provided as a Source Data file.

heterogeneity within psychosis, and indistinguishable from HC in ASD, MDD and ADHD. This study helps moving forward from single modality, ROI-based associations with PRS to multimodal and the whole brain data-driven analysis by both cross cohorts' prediction and classification validation. All above findings indicate the potential of the identified PRS-associated multimodal frontotemporal network to serve as a trans-diagnostic gene intermediated brain signature specific to SZ, which underlies the genetic pathophysiology of schizophrenia and also provide the first evidence for its potential to be used as a biomarker.

## Methods

### The discovery UKB cohort

Ethical approval was obtained from the Human Biology Research Ethics Committee, University of Cambridge (Cambridge, UK). Informed consent was provided by all participants (https://biobank. ctsu.ox.ac.uk/crystal/field.cgi?id=200). The UKB cohort formed the basis of our analyses. This study is under Application ID 34175: Identify biomarkers for distinguishing different mental disorders using brain images and their associations with genetic risk. Discovery participants were recruited from the UK Biobank, a population-based cohort of over 500,000 individuals aged 39–73 years from 22 centers across the United Kingdom between 2006 and 2010. Our study focused on a subset of $N = 22,459$ healthy white people for each of whom completed the genotype and multimodal MRI data[57]. Subjects that have any ICD-10 coded neurological or psychiatric diseases, congenital neurological diseases, that reported themselves that they were told to have a specific neurological/psychiatric disease (which may or may not have been ICD-coded), non-white, with incomplete MRI data were excluded. This is the strictest version of selecting European ancestry unaffected individuals in UKB.

Resting state fMRI (rs-fMRI), T1 (sMRI) and single nucleotide polymorphism (SNP) data were downloaded from UK Biobank[57,58]

(https://biobank.ctsu.ox.ac.uk/crystal/crystal/docs/brain_mri.pdf) and further processed as fALFF and GMV by SPM12 for fusion input. Age, gender, head motion (mean framewise displacement) and site (Manchester, Newcastle and Reading) were regressed out from fALFF/GMV matrices prior to fusion analysis. Details on multimodal imaging parameters and preprocessing pipeline for all the cohorts included in this study can be found in Supplementary "Multimodal imaging parameters and preprocessing" and Supplementary Table 9.

### Independent SZ cohorts

Four independent SZ datasets were included as validation cohorts in this study. SZ and HC were recruited from BSNIP-1, fBIRN, COBRE and MPRC with HC had no current or past history of other psychiatric or neurological illness. Rs-fMRI, sMRI, and SNP were available for all the 4 cohorts, while cognition and symptom were available for 3 cohorts except for MPRC. Different cognitive batteries were used, BSNIP-1: BACS; fBIRN: CMINDS; COBRE: MCCB, as shown in Supplementary Tables 10–12 (details can be found in Supplementary "Cognitive measures").

### Other disorder cohorts

ASD ($N = 421$) and HC ($N = 389$) multimodal data were obtained from the Autism Brain Imaging Data Exchange (ABIDE II)[59]. MDDs ($N = 260$) and HC ($N = 281$) were recruited from Beijing Anding Hospital, West China Hospital of Sichuan, First Affiliated Hospital of Zhejiang and Henan Mental Hospital of Xinxiang[60]. ADHD ($N = 346$) and HC ($N = 478$) data (http://fcon_1000.projects.nitrc.org/indi/adhd200/index.html) were obtained from the ADHD-200 project[61]. Diagnosis of ASD, MDD and ADHD were based on Structured Clinical Interview for DSM-IV. Demographic information of each diagnosed group can be found in Table 1. Rs-fMRI and sMRI were available for ASD, MDD and ADHD cohorts. The same MRI preprocessing pipeline was used to generate fALFF and GMV for UKB, BSNIP-1, fBIRN, COBRE, MPRC, ASD, MDD, and ADHD.

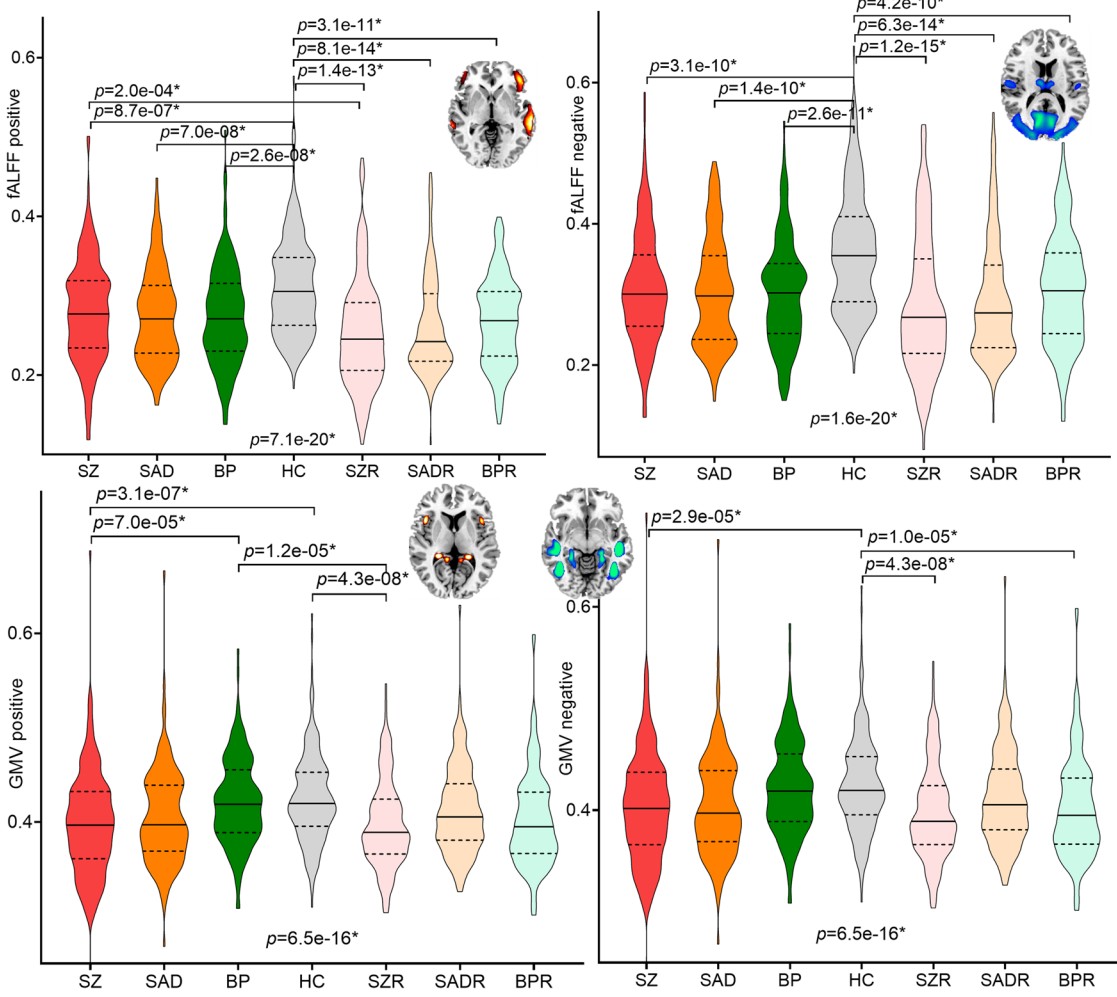

**Fig. 6 | Group differences of the identified PRS-associated pattern within psychosis.** Group differences of the positive and negative PRS-associated networks among SZ, schizo-affective disorder (SAD), psychotic bipolar disorder (BP), HC, schizophrenia relatives (SZR), schizo-affective disorder relatives (SADR) and bipolar disorder relatives (BPR) in BSNIP-1 cohort. Two-tailed two-sample T test was used to calculate the group differences between any two groups within psychosis in Fig. 6.

## PRS scores

For the UKB SNP data, we conducted PLINK quality control (QC) on matched imaging-genetic samples to remove SNPs with missing rate greater than 0.05, SNPs with minor allele frequency less than 0.01, and SNPs in Hardy-Weinberg disequilibrium ($p < 1e−06$). We then conducted sample-level QC using the corresponding genotype data, where we removed samples with high heterozygosity (>3 SD, standard deviation), samples with genotype missing rate greater than 0.05. Multidimensional scaling (MDS) analysis further identified European Ancestry in a strict sense, defined as within 3 SD of the center of 1000G EUR samples. The COBRE, fBIRN, BSNIP-1 and MPRC data were imputed and preprocessed as described in[62]. In brief, we did pre-phasing using SHAPEIT[63], and used IMPUTE2 for imputation[64], with the 1000 Genomes data serving as the reference panel[65]. The imputed data were then QC'ed at INFO score >0.95, and then went through the same PLINK QC as the UKB data. The common SNPs covered by both UKB and COBRE + fBIRN + BSNIP − 1 + MPRC data were obtained, from which we extracted SNPs located in 108 schizophrenia risk loci (PGC SZ 2)[7]. Clump-based linkage disequilibrium pruning was performed with an $r^2 < 0.1$ within a 200-kb window. The PRS for each participant was calculated using PRSice[66] by summing the risk loci weighted (natural log of the odds ratio) by the strength of the association of each SNP with schizophrenia. Six scores were generated using SNPs thresholded at $P_{SNP} ≤ 5.0e−08$, $1.0e−04$, and 0.05 and pruned at $r^2 < 0.1$ and 0.2. $P_{SNP} ≤ 5.0e−08$ with

$r^2 < 0.1$ based on PGC 2 108 loci were used for the main analysis, as these threshold were the most strict GWAS thresholds. The same procedure was used to calculate PRS based on PGC SZ 3 (270 loci, the preprint version)[4].

## Motion and covariates regression

Subjects with mean FD exceeding 1 mm, and head motion exceeding 2.5 mm of maximal translation (in any direction of x, y or z) or 2.5° of maximal rotation throughout the course of scanning were excluded. We further regressed out six head motion parameters (3 translations and 3 rotations) cerebrospinal fluid [CSF] + white matter [WM] + global signal in resting state fMRI preprocessing. The correlation between mean FD and PRS was not significant ($p > 0.05$, Supplementary Table 7). Details can be found in Supplementary "Motion effect on PRS-MRI pattern" section. In addition, mean FD, site, gender and age were regressed out from fALFF/GMV feature matrices prior to the primary fusion analysis.

## Study design

The study design of developing, testing and validating of the PRS-associated multimodal biomarkers was displayed in Fig. 1. Firstly, schizophrenia PRS-guided fusion was performed in healthy white people in UKB. Specifically, subject-wise PRS values were used as a reference to jointly decompose the preprocessed fALFF ($X_1$) and GMV ($X_2$) by "MCCAR + jICA"[61,67,68] to investigate PRS-associated multimodal

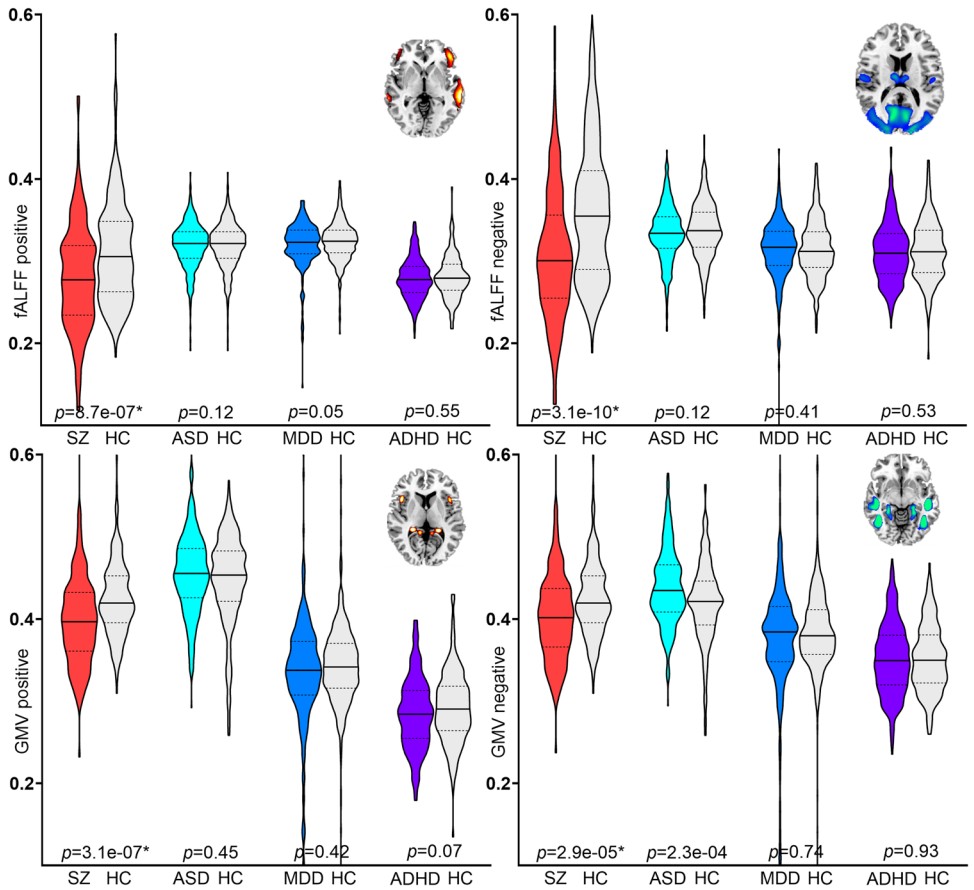

**Fig. 7 | Group differences of the positive and negative PRS-associated networks between SZ and HC, between ASD and HC, between MDD and HC, between ADHD and HC.** The identified PRS-associated features appear to be specific to SZ.

ASD autism spectrum disorder, MDD major depressive disorder, ADHD attention-deficit/hyperactivity disorder. Two-tailed two-sample T test was used to calculate the group differences between patients and controls in Fig. 7.

brain network. The correlations of imaging components with PRS was maximized in the supervised fusion method, as in Eq. 1.

$$\max \sum_{k,j=1}^{2} \left\{ |\mathrm{corr}(\boldsymbol{A}_k, \boldsymbol{A}_j)|_2^2 + 2\lambda \cdot |\mathrm{corr}(\boldsymbol{A}_k, \mathrm{PRS})|_2^2 \right\} \quad (1)$$

where $\boldsymbol{A}_k$ is the loading matrix for each modality, $\mathrm{corr}(\boldsymbol{A}_k, \boldsymbol{A}_j)$ is the column-wise correlation between $\boldsymbol{A}_k$ and $\boldsymbol{A}_j$, and $\mathrm{corr}(\boldsymbol{A}_k, \mathrm{PRS})$ is the column-wise correlation between $\boldsymbol{A}_k$ and PRS. This supervised fusion method can extract a joint multimodal component(s) that significantly associated with PRS.

Then the same PRS-guided fusion was performed on healthy white people ($N = 22,495$), healthy subjects ($N = 24,773$), and all available (pass MRI QC, $N = 37,347$) subjects in UKB with PRS (PGC SZ 2: 108 loci[7]) thresholded at $P_{SNP} < 5.0e{-}08$, $P_{SNP} < 1.0e{-}04$, $P_{SNP} < 0.05$, and pruned at $r^2 < 0.1$ and $r^2 < 0.2$, respectively, to validate the replication of the identified PRS-associated pattern within UKB. PRS-guided fusion was also performed by calculating PRS from the preprint PGC SZ 3: 270 loci[4] to test the similarity of the PRS-pattern between 108 loci and 270 loci. Note that PGC SZ 2:108 loci, $P_{SNP} < 5.0e{-}08$ and $r^2 < 0.1$ were used for the main analysis. The identified PRS-associated multimodal components were then separated as positive ($Z > 0$) and negative ($Z < 0$) brain regions based on the Z-scored brain maps. Two brain masks for each of the two modalities were obtained, i.e., each modality had both positive and negative brain networks (four brain imaging networks in total, Fig. 1b). These masks were then used as ROIs to extract features from every subject for the corresponding modality. The mean values within the identified ROIs were calculated for each subject, generating a $N_{subj} \times 4$ feature vector for the two modalities. For non-UKB cohorts,

the brain ROIs identified from UKB were directly applied to patients' rs-fMRI and sMRI to generate the 4-dimension features.

Two-sample t-tests were used to calculate the group differences between SZ and HC of the identified PRS-associated features in 4 independent SZ cohorts (including BSNIP-1, COBRE, fBIRN and MPRC). Group differences (ANOVA and two-sample t-tests) between SZ and psychosis and their relatives, and between other disorders (ASD, MDD, ADHD) and HC were also tested to show the specificity of PRS pattern to SZ. Note that for the non-UKB patient cohorts, sites were regressed out prior to statistical and predictive analysis.

In the prediction analysis, each of the 4 vectors was normalized to mean = 0, std = 1 to avoid contribution bias in prediction. These features were treated as the linear regressors, and the symptom/cognitive scores were treated as the targeted measures. Multiple linear regression models (Eq. 2) were trained to predict cognition and symptoms for single site COBRE SZ cohort. The same predictive models for each domain and the same ROIs were generalized to predict the corresponding cognitive and symptomatic scores for BSNIP-1 and fBIRN cohorts (MPRC was not included in the prediction analysis since the related clinical measures are not available). Pearson correlations between the true and the predicted values were calculated to validated the generalization and predictability of PRS-associated brain features.

$$\begin{aligned}\text{Predicted scores} = \beta_0 &+ \text{fALFF\_positive} \times \beta_1 + \text{fALFF\_negative} \times \beta_2 \\ &+ \text{GMV\_positive} \times \beta_3 + \text{GMV\_negative} \times \beta_4\end{aligned}$$

$$(2)$$

Apart from the mean $N_{subj} \times 4$ feature vector, the first 5 PCs generating from PCA for each network were also included in classifying SZ

and HC. Note that the first 5 PCs captured 99% variance from the identified PRS-associated ROIs, while the mean only represented <50% variance (Supplementary Fig. 13). A feature matrix in dimension of $N_{subj} \times 24$ in total for PRS-associated multimodal brain features were obtained for the classification analysis. A linear SVM classifier was used to classify SZ and HC based on $N_{subj} \times 24$ PRS-associated brain features across 4 independent SZ cohorts (including BSNIP-1, COBRE, fBIRN, and MPRC). Details can be found in Supplementary "Feature selection and classification".

## Reporting summary
Further information on research design is available in the Nature Research Reporting Summary linked to this article.

## Data availability
The UKB, ADHD, and ASD multimodal data used in the present study can be accessed upon application from UKB, ADHD-200 and ABIDE consortiums. The FBIRN, BSNIP-1, COBRE, MPRC, and MDD data are protected and are not publicly available due to data privacy. The fBIRN, BSNIP-1, COBRE, and MPRC data can be accessed upon reasonable request to V.D.C. (vcalhoun@gsu.edu). The MDD data can be accessed upon request to J.S. Source data are provided with this paper.

## Code availability
The fusion code of "MCCAR+jICA" can be downloaded and used directly by users worldwide, which has been released and integrated in the Fusion ICA Toolbox (FIT, https://trendscenter.org/software/fit).

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

## Acknowledgements

This work was supported by the National Natural Science Foundation of China (62136004, 61876082 and 61732006 [to D.Z.], 82022035, 61773380 [to J.S.]), the Natural Science Foundation of Jiangsu Province, China (BK20220889 [to S.Q.]), the National Key R&D Program of China (2018YFC2001600 and 2018YFC2001602 [to D.Z.]), the National Institute of Health grants (R01EB005846, R01MH117107 and P20GM103472 [to V.D.C.]), and the National Science Foundation (2112455 [to V.D.C.]). We would like to thank Daniel H. Mathalon, Judith M. Ford, James Voyvodic, Bryon A. Mueller, Aysenil Belger, Sarah McEwen, Steven G. Potkin and Adrian Preda for sharing the fBIRN multimodal imaging data.

## Author contributions

S.Q. conceptualized the study, performed the data analysis and wrote the paper. J.C. preprocessed the gene data and calculated the PRS scores for UKB, BSNIP-1, fBIRN, COBRE and MPRC. Z.F. preprocessed the fMRI and sMRI data for UKB, BSNIP-1, fBRIN, COBRE, MPRC, ABIDE II, MDDs, ADHD-200. V.D.C., J.S., G.P., J.B., N.I.P.B., P.K., J.A.T., and D.Z. revised the paper. Y.D. submitted the MRI data application to UKB. J.L. submitted the genetic data application to UKB. X.Y., W.S., and R.J. contributed to the results interpretation and discussion.

## Competing interests

The authors declare no competing interests.
