## [Peer Review File · Nature Communications]

Derivation and utility of schizophrenia polygenic risk associated multimodal MRI frontotemporal networkReviewers' comments:

Reviewer #1 (Remarks to the Author):

This manuscript derives a set of brain imaging measures associated with schizophrenia polygenic risk (PRS) and attempts to validate these brain measures as potential biomarkers of schizophrenia. The paper has several strengths, including the use of large, multi-site datasets for cross-validation and replication. The replicated results in fig 5 are particularly impressive. However, several important controls are missing and the underlying conceptual rationale driving this work is somewhat unclear.

Rationale

Several points in the manuscript mention the goal of validating a biomarker, but a biomarker of what? Much of the validation is based on classifying patients and controls, which a clinician can do easily. Some evidence is provided that the brain measures are sensitive to genetic risk for schizophrenia, but this can also be established by taking a family history. A clinically useful biomarker will provide some predictive information about outcome (e.g., who will get ill, who will respond to treatment, etc), and this is acknowledged in the introduction, but these questions are not addressed here. Greater clarity about precisely what the end goal of this work is would help to frame the paper.

Controls

It seems like site effects were removed from the data via simple linear regression, but it has been shown that this is often inadequate. The authors should perhaps consider a more sophisticated approach, such as ComBat. In the least, the authors should present analyses to show that there are no residual site effects (e.g., results of attempts to classify scan site).

The fMRI data were processed with a fairly minimal processing pipeline, which has been shown to perform poorly in removing motion-related artifact (e.g., Ciric et al. *Neuroimage*, 2017; Parkes et al, *Neuroimage*, 2018). The exclusion criteria are also fairly liberal and do not account for the impact of small head movements on the order of 0.2 mm, which can impact fMRI measures (e.g., Power et al; Satterthwaite et al). A more thorough investigation of the impact of motion on the differences between patients and controls is required.

The imaging phenotypes are quite coarse, averaging over large patches of anatomically distributed brain tissue. This raises the question of whether the classifiers are simply detecting differences in global aspects of brain dysfunction rather than anything specific to the regions considered here. In this regard, the specificity analysis, which focuses just on seeing whether brain regions that correlated with a permuted PRS vector are similar to the observed pattern, do not go far enough. The MRI patterns associated with the permuted vectors should be passed forward to the classification and other analyses to establish a null benchmark for assessing the specificity of the findings to the observed spatial pattern. Demonstrating this would provide strong evidence for the specificity of the results.

It is unclear why the additional 5 PCs are used in the SVM analysis. The mean is being proposed as the biomarker, not the PCs. It seems odd to construct a putative biomarker but then classify using additional features. And why 5 PCs? Furthermore, the mean and the PCs will not be independent (the 1st PC is generally very similar to the mean).

Other

The original analysis defining the biomarker is associated with an extraordinarily low correlation of $r=0.074$. It is significant because of the large sample size, but few people would consider an effect explaining <1% of the variance as noteworthy. Figure 2b shows no association at all. It is therefore remarkable that the associated brain measures have predictive utility in the clinical cohorts. One explanation may be that the clinical analyses are just picking about global aspects of brain pathology, which could be addressed through the specificity analysis mentioned above. Should this not be the case, how do the authors explain the utility of something derived from such a weak association in the discovery cohort?

The results section should provide more detail about the methods as when reporting results.

I could not find much information about the fusion approach at all.

It's unclear which specific p-threshold for selecting SNPs from the original GWAS was used in the PRS computation. Was the same one used throughout for all analyses?

There are grammatical errors throughout. Please check. E.g., Line 63 – should be studies; Line 66 – should be 'represents'... and several others elsewhere.

Reviewer #2 (Remarks to the Author):

Summary: NCOMMS-21-43466 describes a large cross-sectional association study between polygenic risk score (PRS) for schizophrenia (SZ) and multi-modal MRI (T1w & resting state fMRI). Authors establish robust associations between PRS and both modes of MRI data in a large, well-powered sample (N > 22,000). The authors further use these MRI profiles as brain features to detect / predict SZ case-control differences in several independent cohorts. Their results suggest that a pattern of SZ-PRS related structure-function alterations are generalisable, performing well as a classifier in several naïve datasets as well associating with cognition and SZ symptom profiles. The authors further show relative disease specificity – predominately linking the feature profile to psychotic disorders, but not mood / neurodevelopment disorders.

Patterns of associations between SZ PRS and MRI data in UK Biobank have been reported before for both T1w (PMID: 34462574) and ALFF modes (PMID: 32514083), making this part of the investigation largely confirmatory / repeated. The merit / novelty of the current investigation is in demonstrating how these PRS – MRI patterns are observable in SZ cases in other datasets. While this is also not the first study to show that (discovery) SZ-related brain effects are present in independent (test) samples (e.g. PMID: 32463560), it is (to my knowledge), the first to show that PRS-brain associations can distinguish SZ case/control differences – which may inform more about risk that SZ associated confounding. There are a number of points that need further exploration and additional clarification.

Line 60 - Reference 1 does not relate to epidemiology observations of SZ prevalence, but to a prior UKBB PRS-SZ brain imaging study.

Line 64 – Accounts of SZ PRS should be to explain overall liability (which I believe is currently a little over 7%, PMID: 25056061), rather than diagnostic variance).

Line 79 – Polygenic risk for SZ has been associated with brain alterations the cite references [21-24], none of which support a relationship between MRI IPDs and SZ-PRS, especially Franke et al., 2016 [23].

Lines 174-185 – I do not think the term 'replicable' is applicable here, as the authors are making the inference on an extended pool of participants, using different parameters. I would suggest the use of a split / independent sample would be more appropriate to make this inference. I would suggest this would be useful either way, to ensure the fusion model / pattern is not overfitting the data.

Line 181 – well replicated (>80%) with different PRS thresholds is misleading – the only thresholds used are various pruning strategies, not P-thresholds (supplementary material). See also query for 405-409.

Line 224 (and throughout manuscript) – There are well-established negative associations between SZ - PRS and intelligence / cognition (e.g. PMID: 34347035). While The PRS-brain features maybe specific to SZ, how much of this variance can be explained by reduced IQ in UKBB participants with high SZ-PRS and lower IQ in the patient samples?

Line 405 - 409 – It was not clear why the authors chose to restrict their SZ PRS to SNPs that surpassed GWAS threshold ($P < 5 * 10^{-8}$) when more liberal P-threshold (e.g $P < 0.05$) explain more variance in case-control status (PMID: 25056061).

Lines 412-414 – I am slightly concerned about the FD threshold used for exclusion criteria of head motion. A mean FD of 2.5mm (nearly 1 voxel) is extremely liberal censoring protocol (typically > 0.2 mm mean FD for rsfMRI, PMID: 29440410)

Lines 416 - mean FD ~ PRS correlation $P < 0.5$, do authors mean $P < 0.05$, as association between Mean FD and PRS0.1 was $p = 0.09$ (supplementary, line 245)?

Below, we shown the reviewers' comments in black and the responses to the comments in blue. For convenience, we have also included the updated text added to the manuscript which is underlined.

Reviewers' comments:

Reviewer #1 (Remarks to the Author):

This manuscript derives a set of brain imaging measures associated with schizophrenia polygenic risk (PRS) and attempts to validate these brain measures as potential biomarkers of schizophrenia. The paper has several strengths, including the use of large, multi-site datasets for cross-validation and replication. The replicated results in fig. 5 are particularly impressive. However, several important controls are missing and the underlying conceptual rationale driving this work is somewhat unclear.

Thank you for the careful reading, helpful comments, and constructive suggestions, which have significantly improved the presentation of the manuscript.

1. Rationale

Several points in the manuscript mention the goal of validating a biomarker, but a biomarker of what? Much of the validation is based on classifying patients and controls, which a clinician can do easily. Some evidence is provided that the brain measures are sensitive to genetic risk for schizophrenia, but this can also be established by taking a family history. A clinically useful biomarker will provide some predictive information about outcome (e.g., who will get ill, who will respond to treatment, etc), and this is acknowledged in the introduction, but these questions are not addressed here. Greater clarity about precisely what the end goal of this work is would help to frame the paper.

Thank you for the comments. Our responses are below:

1.1 “a biomarker of what?”

We identified a schizophrenia polygenic risk (PRS)-associated multimodal frontotemporal brain network based on large UKB dataset (healthy Caucasian, N = 22,459), *i.e.*, **a MRI biomarker of schizophrenic genetic risk**. All previous PRS and imaging associating analysis were ROI-based single modality correlation analysis without demonstrating the biomarker ability (neither classification nor prediction). In contrast, this study first investigated schizophrenic PRS-associated pattern using fALFF and GMV, and further verified its diagnostic ability in discriminating SZ from HC and its predictive ability in estimating the risk scale, symptom severity and cognitive scores for SZ in four independent SZ cohorts. Its specificity for SZ was also demonstrated by multi-disease comparison: most severe in schizophrenia and schizo-affective disorder, milder in bipolar disorder, and indistinguishable from

healthy controls in autism, depression and attention-deficit hyperactivity disorder.

1.2 “Much of the validation is based on classifying patients and controls, which a clinician can do easily.”

We acknowledge that a psychiatrist could diagnose who has schizophrenia. However, the problem is that there are no existing gold standards for the diagnoses of schizophrenia. The current diagnostic systems (DSM-V and ICD-10) for psychiatry mainly rely upon presented signs and self-reported symptoms. In other words, the diagnosis is based on a descriptive collection of behaviors without the availability of any objective test to diagnose patients. Thus DSM/ICD diagnoses are highly reliable without underlying biological validity. And, perhaps most important, the presenting signs and symptoms, do not capture fundamental underlying mechanisms of neuro-dysfunction for schizophrenia (identical symptoms with polar opposite mechanisms). Indeed, the Research Domain Criteria (RDoC) within the US NIMH were proposed to help move towards a diagnostic system that has both reliability and validity by identifying more **objective** biomarkers to diagnose mental disorders¹. The test-retest and interrater reliability for SZ diagnosis is 0.8 [95% CI: 0.76–0.84], however for schizo-affective disorder this is only 0.57 [95% CI: 0.41-0.73]². The multimodal MRI-based machine learning pipeline proposed in this study provides a SZ diagnosis reliability at 0.86 [95% CI: 0.84–0.9]. Additional MRI biomarker identification might **increase diagnostic reliability**³ and inform treatment decisions. For these patients, a resting state fMRI and sMRI analysis might be an important diagnostic improvement informing treatment decisions (note that only MRI features were used in the classification and prediction analyses, while the PRS was not included). The fact that the reported classification and prediction were implemented with data obtained from rest-fMRI and sMRI scans of **under 10 minutes’ duration**, speaks to the feasibility and cost-effectiveness of our approach (**clinician needs at least 15 minutes’ clinical interview** to diagnose patients). Furthermore, the identified PRS-related pattern that might assist the clinician in differentiating whether a first episode psychotic patient has schizophrenia versus bipolar-I with psychosis, would be clinically useful. Also, assistance in differentiating prodromal SZ in patients with Autism Spectrum Disorder who are exhibiting odd beliefs or perceptions, could be important.

Collectively, identifying neuroimaging features to classify patients and controls will contribute to identify an objective dimension that can achieve high diagnostic reliability and consistency across different raters and also provide potentially important biological information about the disorder to validate categories.

1.3 “Greater clarity about precisely what the end goal of this work is would help to frame the paper.”

We appreciate this point. The end goal of this study is to 1) identify the genetic risk-associated MRI features in schizophrenia, and 2) whether these features can be used to distinguish individuals with and without the disorder, and 3) whether those same features are related to cognitive deficits and overall symptom severity, and further 4) whether such imaging features are specific to schizophrenia. Although brain abnormalities linked with PRS have been reported in schizophrenia, prior studies focused on a single imaging modality and used a region of interest (ROI) based simple correlation analysis. Through a novel multimodal fusion with reference approach, we are able to identify PRS-associated brain signatures, which may serve as potential objective biomarkers for disease identification, risk prediction, and symptom estimation after rigorous cross validation.

To clarify, the existing literature in the field of PRS and neuroimaging association studies in SZ has mainly focused on pathophysiology explanations of SZ, while our current study goes beyond PRS and neuroimaging associations by testing its ability in classification and prediction, and demonstrating its specificity within psychosis (frontotemporal alterations were most severe in schizophrenia and schizo-affective, milder in bipolar), and indistinguishable from HC in autism, depression and ADHD.

In conclusion, this work moves the field forward from unimodal, ROI-based associations with PRS to multimodal, whole-brain data-driven analysis with stringent cross-cohort validation in prediction and classification. **These findings indicate the potential of the identified PRS-associated multimodal frontotemporal network to serve as a trans-diagnostic gene intermediate brain signature specific to SZ, which underlies the genetic pathophysiology of schizophrenia and also provides the first evidence for its potential to be used as a biomarker.**

We have revised the introduction and discussion according to the reviewer’s comments, see page 4 and 20-23. “Biomarker” was removed from the title, and was replaced by “potential biomarker” throughout the manuscript.

2. Controls

It seems like site effects were removed from the data via simple linear regression, but it has been shown that this is often inadequate. The authors should perhaps consider a more sophisticated approach, such as ComBat. In the least, the authors should present analyses to show that there are no residual site effects (e.g., results of attempts to classify scan site).

Thank you for the valuable comments. As mention in ⁴, ComBat substantially increase the statistical significance of neuroimaging findings compared to random-effects meta-analyses. However, it was proposed to reduce the heterogeneity between sites related factors in meta-analysis generating

from different sites (such as ENIGMA project). For ENIGMA, researchers using the same pipeline to perform the neuroimaging preprocessing, then share the statistical results to do the meta/mega analysis. This means that the researchers do not actually share the data, but only share the results to derive a pooled estimation of statistical significance. However, in current study, we did have imaging data from each cohort. Here, in response to the reviewer, we added several additional experiments with regard to discuss the site effect.

2.1 Site effect on PRS-MRI fusion within UKB

For the MRI imaging data, there are three sites available in UKB, including Cheadle, Reading and Newcastle. We performed the PRS-guided fusion within each site separately to test the similarity of the identified PRS-associated frontotemporal pattern. Dice index, equation (1) was used to calculate the overlap percentage of the spatial maps between sites. Dice index is a statistical validation for comparing the spatial similarity of binary images, for example in image segmentation accuracy assessment. We calculated the Dice index of the identified PRS-associated component between two cohorts using only voxels masked at $|Z|>2$, resulting in two masks from UKB (mask_UKB) and Cheadle/Reading/Newcastle (mask_Cheadle/Reading/Newcastle) respectively. Only voxels that fell into the union of the masks ($\text{mask_UKB} \cup \text{mask_Cheadle}$) were used to calculate the cross-cohort similarity as shown in equation (1).

$$\text{Dice index} = 2 \frac{V(A \cap B)}{V(A) + V(B)} \quad (1)$$

Supplementary Table 4. Spatial similarity between UKB and Cheadle/Reading/Newcastle sites.

Dice	Cheadle	Reading	Newcastle
UKB fALFF	0.92	0.75	0.77
UKB GMV	0.91	0.70	0.81

Site effects on SZ PRS associated multimodal patterns

(a) UKB (N=22773)

(b) Cheadle (N=13012)

(c) Reading (N=3398)

(d) Newcastle (N=6049)

Supplementary Figure 9. PRS pattern for UKB (a), Cheadle (b), Reading (c), and Newcastle (d). The spatial similarities were > 0.70 across UKB, Cheadle, Reading, and Newcastle for both fALFF and GMV components (**Supplementary Table 4**).

The above results (**Supplementary Table 4** and **Supplementary Fig. 9**) showed that there were high spatial similarities among Cheadle, Reading, Newcastle and UKB. The Dice index for GMV and fALFF components were all > 0.70 , suggesting that there was high overlap percentage of the spatial maps cross different sites within UKB. Thus we do not believe that site would be a major confounding factor with respect to the identified PRS frontotemporal multimodal pattern.

2.2 Site effect in classification

There are 4 independent SZ cohorts (BSNIP, COBRE, fBIRN, MPRC) included in our current study. However, different SZ cohorts consist different number of sites. There are 5 sites for BSNIP, 1 site for COBRE, 7 sites for fBIRN and 3 sites for MPRC. Since COBRE is a single site, so the

classification on scanning sites are performed for BSNIP (class=3), fBIRN (class=7) and MPRC (class=3). The mean fALFF/GMV plus the first 5PCs within positive and negative PRS-associated brain networks were used as feature input and sites were treated as labels in the SVM classifications. Results (**Supplementary Fig. 14**) showed that all the classification accuracies were approximated as around 50% as a random distributed accuracy (the more number of site the lower classification accuracy). **This means that site is not a major confounding factor for the current SZ-HC classification results.**

Supplementary Figure 14. The classification results on scan sites for BSNIP, fBIRN and MPRC cohorts. Upper row represents ROC; lower row represents confusion matrix.

2.3 Site effect on PRS, PANSS and cognition

ANOVA test (site was used as covariate) showed that there was no site difference of PRS ($p = 0.96$) within UKB. The site differences of PANSS and cognition for independent SZ cohorts were shown in the following Table. Since clinical scores are not available for MPRC cohort, and COBRE is a single site cohort, so these two cohorts were not included in the following Table.

Site differences on cognition and PANSS scores for fBRIN and BSNIP cohorts.

Anova (p value)	Cognition	PANSS positive	PANSS negative
BSNIP	0.13	0.093	0.165
fBIRN	0.08	0.93	0.86

Collectively, all above results indicate that **site is not a major confounding factor for the identified PRS pattern and SZ-HC classification**. We have added these contents to **Supplementary “Site effect”** section.

3. The fMRI data were processed with a fairly minimal processing pipeline, which has been shown to perform poorly in removing motion-related artifact (e.g., Ciric et al. Neuroimage, 2017; Parkes et al, Neuroimage, 2018). The exclusion criteria are also fairly liberal and do not account for the impact of small head movements on the order of 0.2 mm, which can impact fMRI measures (e.g., Power et al; Satterthwaite et al). A more thorough investigation of the impact of motion on the differences between patients and controls is required.

Thank you for the comments. We have added the following experiments with regard to the effect of head motion.

3.1 Motion on preprocessing

Please note that in the preprocessing procedure for fMRI, we despiked the fMRI data: nuisance covariates (6 head motions + cerebrospinal fluid [CSF] + white matter [WM]) + global signal were regressed out via a general linear model from the voxel time series. We did remove outlier subjects who have framewise displacements (FD) exceeding 1.0 mm, as well as head motion exceeding 2.5 mm of maximal translation (in any direction of x, y or z) or 1.0° of maximal rotation throughout the course of scanning. Results indicate all FDs (mean framewise displacements, mean of root of mean square frame-to-frame head motions assuming 50 mm head radius⁵) for all subjects were <1 mm at every time point. In addition to denoising the BOLD fMRI signal in preprocessing, ICA approach used in current study can further isolate artifact components (motion and noise) from fMRI⁶.

3.2 PRS pattern within UKB subset with head motion <0.2mm

As requested from the reviewer, we also excluded subjects with >0.2mm FD to get a subset of UKB (N = 13490, 60% subjects’ head motion <0.2mm) to perform the same PRS-guided fusion to test

whether the identified multimodal frontotemporal pattern can be validated. Result (**Supplementary Fig. 10**) showed that the identified PRS-associated pattern (frontotemporal cortex and thalamus in fALFF, accompanied with thalamus, hippocampus, para-hippocampus and temporal cortex in GMV) can be replicated on UKB subset with $FD < 0.2\text{mm}$. This tells us that the head motion is not a major confounding factor for the current fusion results.

Supplementary Figure 10. (a) The original PRS-associated fALFF+GMV covarying pattern for UKB. (b) The PRS pattern on UKB subset with head motion < 0.2mm.

3.3 Group differences of mean FD between SZ and HC

We have calculated the group differences of mean FD between HC and SZ across the 4 SZ cohorts included in this study. Note that there were no significant differences between patients and controls on mean FD for all the 4 SZ cohorts, namely,

- BSNIP, HC: mean= $0.22 \pm 0.11\text{mm}$, SZ: $0.26 \pm 0.24\text{mm}$, two sample t-test: $p = 0.25$
- FBIRN, HC: mean= $0.25 \pm 0.18\text{mm}$, SZ: $0.27 \pm 0.21\text{mm}$, two sample t-test: $p = 0.65$
- COBRE, HC: mean= $0.22 \pm 0.12\text{mm}$, SZ: $0.21 \pm 0.11\text{mm}$, two sample t-test: $p = 0.77$
- MPRC, HC: mean= $0.12 \pm 0.22\text{mm}$, SZ: $0.18 \pm 0.10\text{mm}$, two sample t-test: $p = 0.15$

3.4 Partial correlation after regressing out mean FD

Partial correlation has been proposed as an alternative approach for removing spurious shared variance in correlation analysis⁷. Here, we also performed partial correlation analysis between the identified component and PRS by regressing out mean FD. Result showed that the significance level was not changed by mean FD ($p = 5.2\text{e-}30^*$ for fALFF, $p = 2.3\text{e-}28^*$ for GMV as in **Fig. 2b**).

3.5 fALFF not functional connectivity

- [1] Satterthwaite TD, Wolf DH, Loughead J, Ruparel K, Elliott MA, Hakonarson H, Gur RC, Gur RE. Impact of in-scanner head motion on multiple measures of functional connectivity: relevance for studies of neurodevelopment in youth. *Neuroimage*. 2012;60:623-632.
- [2] Satterthwaite TD, Elliott MA, Gerraty RT, Ruparel K, Loughead J, Calkins ME, Eickhoff SB, Hakonarson H, Gur RC, Gur RE, Wolf DH. An improved framework for confound regression and filtering for control of motion artifact in the preprocessing of resting-state functional connectivity data. *Neuroimage*. 2013;64:240-256.
- [3] Ciric R, Wolf DH, Power JD, Roalf DR, Baum GL, Ruparel K, Shinohara RT, Elliott MA, Eickhoff SB, Davatzikos C, Gur RC, Gur RE, Bassett DS, Satterthwaite TD. Benchmarking of participant-level confound regression strategies for the control of motion artifact in studies of functional connectivity. *Neuroimage*. 2017;154:174-187.
- [4] Ciric R, Rosen AFG, Erus G, Cieslak M, Adebimpe A, Cook PA, Bassett DS, Davatzikos C, Wolf DH, Satterthwaite TD. Mitigating head motion artifact in functional connectivity MRI. *Nature protocols*. 2018;13:2801-2826.
- [5] Power JD, Plitt M, Gotts SJ, Kundu P, Voon V, Bandettini PA, Martin A. Ridding fMRI data of motion-related influences: Removal of signals with distinct spatial and physical bases in multiecho data. *Proceedings of the National Academy of Sciences of the United States of America*. 2018;115:E2105-E2114.

The above recommended papers found that head motion was sensitive to functional connectivity analysis. However, the current fusion analysis was conducted on the spatial maps of fALFF not functional connectivity. While it is the functional connectivity derived from rs-fMRI that is more sensitive to head motion⁸⁻¹².

Collectively, considering there was no group difference in head motion between SZ and HC, and no significant correlation between mean FD and PRS, and the partial correlation between the identified component and PRS remained significant after regressing out mean FD, the PRS-pattern can be replicated within UKB subset with head motion < 0.2mm, and the current fusion analysis was based on fALFF not functional connectivity, we believe that micro-motion is not a major factor affecting the current results.

The motion related papers referred by the reviewer were cited in the revised manuscript, and the above contents were added in **Supplementary “Motion effect”** section and the main text **“Site and motion effects on the identified PRS-pattern”** section (page 12-13), respectively.

4. The imaging phenotypes are quite coarse, averaging over large patches of anatomically distributed brain tissue. This raises the question of whether the classifiers are simply detecting differences in global aspects of brain dysfunction rather than anything specific to the regions considered here. In this regard, the specificity analysis, which focuses just on seeing whether brain regions that correlated with a permuted PRS vector are similar to the observed pattern, do not go far enough. The MRI patterns

associated with the permuted vectors should be passed forward to the classification and other analyses to establish a null benchmark for assessing the specificity of the findings to the observed spatial pattern. Demonstrating this would provide strong evidence for the specificity of the results.

Thank you for the instructive comments. Here, in response to the reviewer, we emphasized the following points:

4.1 Imaging phenotypes

“The imaging phenotypes are quite coarse, averaging over large patches of anatomically distributed brain tissue.”

The multimodal imaging phenotypes (fALFF and GMV) used in the current study are voxel-wise measures, not averaging over large patches. To calculate fractional amplitude of low frequency fluctuations (fALFF)¹³, the sum of the amplitude values in the 0.01 to 0.08 Hz low-frequency power range was divided by the sum of the amplitudes over the entire detectable power spectrum (range: 0–0.25 Hz)¹⁴. GMV was generated from the segmented sMRI. Both fALFF and GMV were calculated voxel-wise.

4.2 Specificity in classification and group difference test

“This raises the question of whether the classifiers are simply detecting differences in global aspects of brain dysfunction rather than anything specific to the regions considered here.”

Note that both the classification and group difference analysis between SZ and HC were based on the features extracted from the identified PRS-associated frontotemporal pattern (targeted ROIs). So it is not the classifiers detecting differences in global brain dysfunction.

“The MRI patterns associated with the permuted vectors should be passed forward to the classification and other analyses to establish a null benchmark for assessing the specificity of the findings to the observed spatial pattern. Demonstrating this would provide strong evidence for the specificity of the results.”

In response to the reviewer, we performed both the classification and group difference test based on features from the null pattern (Supplementary Fig. 4b) identified from the permuted PRS. The same feature extraction procedure was used to generate features from the null pattern (the mean plus the first 5PCs from the identified ROIs). As displayed in Supplementary Fig. 13, there are no group differences between SZ and HC of the null pattern, and the classification accuracy is around 50% across the 4 independent SZ cohorts. When comparing with Fig. 4 (the main text), it is clear that the

identified PRS-associated frontotemporal pattern is specific in discriminating between SZ and HC.

Supplementary Figure 4. (a) The fALFF+GMV covarying pattern associated with PRS. (b) The most frequently occurring (voxels with more than 60% occurrences) covarying pattern associated with 500 times permuted PRS.

Supplementary Figure 13. (a) Group differences between SZ and HC of the null pattern (Supplementary Fig. 4b) for independent BSNIP-1, COBRE, fBIRN and MPRC cohorts, respectively. (b) ROC curves of the classification results between SZ and HC for BSNIP-1, COBRE, fBIRN and MPRC cohorts, respectively.

Figure 4. (a) Group differences between SZ and HC of PRS-associated the positive and negative networks for independent BSNIP-1, COBRE, fBIRN and MPRC cohorts, respectively. (b) ROC curves of the classification results between SZ and HC for BSNIP-1, COBRE, fBIRN and MPRC cohorts, respectively.

The above contents on the specificity demonstration of the identified PRS-associated pattern in classification and group difference analysis were added to the main text (page 13-14) and Supplementary file.

5. It is unclear why the additional 5 PCs are used in the SVM analysis. The mean is being proposed as the biomarker, not the PCs. It seems odd to construct a putative biomarker but then classify using additional features. And why 5 PCs? Furthermore, the mean and the PCs will not be independent (the 1st PC is generally very similar to the mean).

Sorry for the lack of clarification, we response to each of these points below.

5.1 “Why additional 5 PCs?”

PCA is a dimensionality-reduction method that is often used to reduce the dimensionality by transforming a large set of variables into a smaller one (linearly uncorrelated, *i.e.*, orthogonal) that still contains most of the information in the data set. Here, we use the COBRE cohort (the single site data) as an example to show the variance explained and the contribution weight in classification for all the features (the mean and the 5 PCs). Note that the **first 5 PCs captured 99% variance from the identified PRS-associated ROIs, while the mean only represented <50% variance (Supplementary Fig. 11)**. In addition, we also plotted the beta weights for the mean and the 5 PCs in differentiating between SZ and HC. It is clear that different features contribute differently to the classification, and the 1st PC contributes the most, followed by the mean (**Supplementary Fig. 12**). Comparing with group difference tests, classification is a more complex task, which means that only the mean feature is not enough (**Supplementary Fig. 11-12**). This is why we choose 5 additional PCs in classification. Moreover, **the 5 PCs were all extracted from the same identified PRS-associated frontotemporal brain ROIs, not from the whole brain**. This tells us that it is the PRS related brain region we identified that is informative for the classification analysis.

Supplementary Figure 11. Percentage of the explained variance comparing PCA components and the mean extracted from the identified PRS-associated frontotemporal ROIs.

Supplementary Figure 12. Beta weights in classification comparing PCA components and the mean extracted from the identified PRS-associated frontotemporal ROIs.

5.2 “the 1st PC is generally very similar to the mean”.

Before PCA, all the variables need to be normalized to the same scale, *i.e.*, standardization. Mathematically, normalization can be done by subtracting the mean and dividing by the standard deviation for each value of each variable (Eq. 1).

$$Z = \frac{\text{value} - \text{mean}}{\text{standard variation}} \quad (1)$$

Then the eigenvalues and eigenvectors were computed from the covariance matrix to extract the principle components. So, under certain conditions the mean can be called "a very crude relative of PCA", in a sense that PCA will result in the first principal component being proportional to the average of all variables (or close to it). **Sphericity is perfect or near-perfect. All variables are highly orthogonal to all other variables and can be uniquely estimated.** Moreover, this often remains approximately true if the off-diagonal elements of the covariance matrix are not exactly equal, but are of similar magnitude. Under these ideally conditions, the first PC will often be close to the average. However, in the current data, not all the variables are perfectly orthogonal to each other, which means that the mean is not similar to the 1st PC. We have also calculated the correlation between the mean and the 5 PCs. The correlation between the mean and 1st PC is not 1 nor close to 1 (**Supplementary Table 6**).

Supplementary Table 6. Correlation between the mean the 5 PCs.

Correlation	1 st	2 nd	3 rd	4 th	5 th
r	0.53	0.02	0.08	0.27	-0.04
p	1.2e-20	0.80	0.25	1.7e-04	0.58

Furthermore, the variance explained by the mean and the 1st PC is not equal (**Supplementary Fig. 11**), neither the contribution weight in classification analysis (**Supplementary Fig. 12**). All the above evidence demonstrate that the mean is not similar to the 1st PC as in the current dataset.

Other

6. The original analysis defining the biomarker is associated with an extraordinarily low correlation of $r=0.074$. It is significant because of the large sample size, but few people would consider an effect explaining <1% of the variance as noteworthy. Figure 2b shows no association at all. It is therefore remarkable that the associated brain measures have predictive utility in the clinical cohorts. One explanation may be that the clinical analyses are just picking about global aspects of brain pathology, which could be addressed through the specificity analysis mentioned above. Should this not be the case, how do the authors explain the utility of something derived from such a weak association in the discovery cohort?

We agree the correlation between the identified IC and PRS is not very high due to the large sample size. However, compared to the existing studies based on UKB, we found that it is normal that the variance explained was <1% by correlating SZ-PRS with imaging phenotypes^{15,16} under different P_{SNP} thresholds and for all the cortical and subcortical areas. This is consistent with a recently published study in Nature 2022 that smaller (sample size) brain wide association studies have reported

larger correlations than the largest effects measured in larger samples¹⁷.

Figure. As reported in ¹⁷, the correlation between cortical thickness and clinical measures decrease as sample size increase (left). And the associations were inflated with small sample size (right).

¹⁷Marek, S., et al. Reproducible brain-wide association studies require thousands of individuals. *Nature* 603, 654-660 (2022).

6.1 Existing PRS-imaging association studies based on UKB

Figure. As reported in ¹⁶, associations between schizophrenia PRS and global cortical and regional

subcortical metrics of human brain structure. **A.** Barcharts of variance explained by schizophrenia PRS (R^2 , y-axis) constructed at each of eight probability thresholds ($0.0001 \leq P_{\text{SNP}} \leq 1$, x-axis) for each of nine global mean cortical metrics: CT cortical thickness, Vol grey matter volume, SA surface area, IC intrinsic curvature, LGI local gyrification index, FA fractional anisotropy, MD mean diffusivity, NDI neurite density index, ODI orientation dispersion index. **B.** Barcharts of variance explained by PRS (R^2 , y-axis) constructed at each of eight probability thresholds ($0.0001 \leq P_{\text{SNP}} \leq 1$, x-axis) for NDI measured at each of seven subcortical regions.

¹⁶Stauffer, E.M., *et al.* Grey and white matter microstructure is associated with polygenic risk for schizophrenia. *Molecular psychiatry* (2021).

¹⁵Neilson, E., *et al.* Impact of Polygenic Risk for Schizophrenia on Cortical Structure in UK Biobank. *Biological psychiatry* **86**, 536-544 (2019).

The above published UKB studies showed that the variance explained was <1% by correlating SZ-PRS with imaging phenotypes^{15,16} under different P_{SNP} thresholds and for all the cortical and subcortical areas.

6.2 For the current study

We have calculated the direct correlation between SZ-PRS and voxel wise MRI features throughout the brain (60758 and 90638 voxels for fALFF and GMV). The maximum absolute correlation r is only 0.03 and 0.028, and the mean r is 0.008 and 0.0006 for fALFF and GMV respectively.

Apart from the voxel wise correlation between SZ-PRS and MRI features, we also tested the correlation between the mean values extracted from ALL atlas and SZ-PRS for both fALFF and GMV under different P_{SNP} thresholds. Results (**Supplementary Fig. 3 and Supplementary Table 1**) showed that the variance explained was <1% for all the brain areas under 3 different P_{SNP} thresholds.

Supplementary Figure 3. Correlations between PRS and the mean values extracted from AAL atlas (90 ROIs) for both fALFF and GMV under different P_{SNP} thresholds (5.0e-08, 1.0e-04, 0.05).

Supplementary Table 1. Direct correlations between SZ PRS and the mean values extracted from AAL atlas (90 areas) for both fALFF and GMV under different P_{SNP} thresholds (5.0e-08, 1.0e-04, 0.05).

Correlation r	$P_{\text{SNP}} = 5.0\text{e-}08$		$P_{\text{SNP}} = 1.0\text{e-}04$		$P_{\text{SNP}} = 0.05$	
	fALFF	GMV	fALFF	GMV	fALFF	GMV
AAL 1	-0.017	-0.017	-0.022	-0.0092	0.010	0.0042
AAL 2	0.011	0.016	-0.018	-0.0090	0.0048	0.0038
AAL 3	0.027	-0.021	0.022	-0.010	0.0027	0.0044
AAL 4	-0.024	0.019	0.026	-0.006	0.0005	0.0048
AAL 5	0.014	0.020	0.013	-0.022	0.0073	-0.0049
AAL 6	-0.012	-0.017	-0.0010	-0.013	0.0031	0.0019
AAL 7	0.028	-0.023	-0.025	-0.012	0.0070	0.00081
AAL 8	-0.021	0.026	0.016	-0.014	0.0037	-0.00068
AAL 9	-0.012	-0.022	0.0073	-0.017	0.0011	-0.0022
AAL 10	-0.0058	-0.018	-0.0013	-0.021	0.00046	-0.0018

AAL 11	-0.020	-0.026	0.017	-0.019	0.0057	-0.0040
AAL 12	-0.014	-0.024	-0.0072	-0.019	0.0071	-0.0022
AAL 13	-0.022	-0.021	0.020	-0.018	0.0027	-0.0039
AAL 14	-0.015	-0.022	-0.0092	-0.022	0.0051	-0.0040
AAL 15	-0.020	-0.021	-0.012	-0.020	0.0049	-0.0065
AAL 16	-0.016	-0.021	-0.006	-0.024	0.0037	-0.00556
AAL 17	-0.0073	-0.017	-0.015	-0.015	-0.0071	-0.00031
AAL 18	-0.0058	-0.021	-0.014	-0.018	-0.0038	-0.0045
AAL 19	-0.012	-0.0081	-0.023	-0.0039	-0.0043	0.0053
AAL 20	-0.015	-0.011	-0.025	-0.0041	-0.0041	0.0079
AAL 21	-0.015	-0.021	-0.0066	-0.019	-0.0059	-0.013
AAL 22	-0.014	-0.027	-0.0097	-0.024	-0.0072	-0.0083
AAL 23	-0.024	-0.024	-0.015	-0.013	0.0011	0.00014
AAL 24	-0.023	-0.024	-0.017	-0.012	0.0027	-0.00033
AAL 25	-0.030	-0.023	-0.023	-0.020	-0.005	-0.0015
AAL 26	-0.024	-0.022	-0.018	-0.016	-0.00068	-0.0017
AAL 27	-0.019	-0.024	-0.0049	-0.021	-0.0073	-0.0045
AAL 28	-0.0070	-0.022	-0.00091	-0.018	-0.0014	-0.0029
AAL 29	-0.013	0.018	-0.015	0.014	-0.0051	0.00096
AAL 30	-0.011	0.017	-0.012	0.016	-0.0022	0.0014
AAL 31	-0.035	-0.018	-0.027	-0.018	-0.012	-0.0024
AAL 32	-0.029	-0.025	-0.026	-0.022	-0.008	-0.0013
AAL 33	-0.017	-0.010	-0.023	-0.013	-0.017	0.00072
AAL 34	-0.021	-0.017	-0.023	-0.016	-0.016	0.0038
AAL 35	-0.020	-0.019	-0.024	-0.016	-0.011	0.0015
AAL 36	-0.024	-0.023	-0.020	-0.016	-0.0072	-0.0035
AAL 37	-0.020	0.014	-0.027	0.014	-0.013	0.0086
AAL 38	-0.014	0.020	0.022	-0.019	-0.011	0.0068
AAL 39	-0.019	0.022	0.022	-0.018	-0.014	0.0026
AAL 40	-0.016	0.022	-0.020	0.017	-0.011	0.0036
AAL 41	-0.019	-0.011	-0.022	-0.016	-0.011	-0.010
AAL 42	-0.0058	-0.017	-0.012	-0.020	-0.011	-0.0058
AAL 43	-0.024	-0.018	-0.023	-0.014	-0.012	0.0039
AAL 44	-0.021	-0.020	-0.022	-0.015	-0.011	-0.001
AAL 45	-0.023	-0.0083	-0.023	-0.0024	-0.012	0.0076
AAL 46	-0.023	-0.017	-0.023	-0.014	-0.015	0.003
AAL 47	-0.018	-0.019	-0.023	-0.013	-0.012	-0.00098
AAL 48	-0.014	-0.020	-0.023	-0.012	-0.014	-0.00068
AAL 49	-0.018	-0.013	-0.021	-0.009	-0.014	0.0013
AAL 50	-0.020	-0.0087	-0.017	-0.010	-0.011	0.0035
AAL 51	-0.024	-0.020	-0.021	-0.015	-0.012	0.00075
AAL 52	-0.027	-0.012	-0.018	-0.010	-0.0099	0.0048
AAL 53	-0.018	-0.015	-0.017	-0.0094	-0.011	-0.0034
AAL 54	-0.017	-0.014	-0.016	-0.011	-0.012	-0.00096
AAL 55	-0.016	-0.018	-0.018	-0.021	-0.012	-0.0021
AAL 56	-0.012	-0.019	-0.017	-0.017	-0.010	-0.0011
AAL 57	-0.0078	-0.016	-0.018	-0.012	-0.015	-0.00015

AAL 58	-0.0075	-0.011	-0.016	-0.015	-0.012	-0.0015
AAL 59	-0.01	-0.013	-0.019	-0.007	-0.0063	0.0021
AAL 60	-0.010	-0.011	-0.016	-0.0011	-0.0020	0.0058
AAL 61	-0.014	-0.016	-0.018	-0.009	-0.0031	0.0019
AAL 62	-0.010	-0.010	-0.0097	-0.00065	-0.0017	0.0002
AAL 63	-0.018	-0.017	-0.015	-0.0099	-0.0044	0.0034
AAL 64	-0.019	-0.020	-0.020	-0.011	-0.0058	-0.0044
AAL 65	-0.019	-0.019	-0.020	-0.013	-0.00051	0.0015
AAL 66	-0.014	-0.018	-0.0078	-0.0073	0.0075	0.0013
AAL 67	-0.020	-0.016	-0.023	-0.0096	-0.0070	0.0033
AAL 68	-0.021	-0.016	-0.020	-0.015	-0.0089	0.0037
AAL 69	-0.0095	-9.6e-05	-0.020	-0.0083	-0.0067	0.0032
AAL 70	-0.013	-0.0041	-0.020	-0.010	-0.011	0.0018
AAL 71	-0.014	-0.013	-0.018	-0.00096	-0.0084	0.0015
AAL 72	-0.012	-0.014	-0.013	-0.00024	-0.0097	0.00665
AAL 73	-0.014	-0.013	-0.017	-0.015	-0.016	-0.0053
AAL 74	-0.0098	-0.016	-0.015	-0.016	-0.010	-0.0027
AAL 74	-0.012	0.00021	-0.015	-0.0033	-0.015	0.0047
AAL 76	-0.0080	-0.0017	-0.010	-0.015	-0.0099	-0.00031
AAL 77	-0.013	-0.014	-0.022	-0.0023	-0.022	-0.0037
AAL 78	-0.011	-0.015	-0.021	-0.0069	-0.019	-0.0046
AAL 79	-0.0059	-0.025	-0.013	-0.022	-0.012	-0.0055
AAL 80	-0.0074	-0.018	-0.010	-0.018	-0.014	-0.00032
AAL 81	0.012	-0.024	0.0088	-0.016	0.011	0.00015
AAL 82	0.010	-0.024	-0.013	-0.016	0.0041	4.9e-05
AAL 83	-0.013	-0.025	-0.0082	-0.019	0.0062	-0.0025
AAL 84	-0.0071	-0.020	0.0060	-0.015	0.00042	-0.0021
AAL 85	-0.011	-0.021	0.012	-0.016	0.00081	0.00042
AAL 86	-0.012	-0.021	-0.010	-0.015	0.0050	0.0023
AAL 87	0.0063	-0.015	0.0022	-0.0096	0.0052	0.0024
AAL 88	-0.00068	-0.014	0.0053	-0.014	0.011	0.0025
AAL 89	0.014	-0.023	-0.011	-0.022	0.0010	-0.003
AAL 90	-0.0096	-0.023	-0.0070	-0.020	0.0041	-0.0019

6.3 Power analysis

In response to the reviewer, we also calculated the statistical power of SZ-PRS and the two MRI features for fusion input (fALFF and GMV) using G*Power software¹⁸ (<http://www.softpedia.com/get/Science-CAD/G-Power.shtml>). As in this study, the sample size is N=22,773 HCs. The effect size of PRS correlates with fALFF loadings is $r = 0.074$. Given the significance level $\alpha = 0.05$, sample size (N=22,773), and the effect size = 0.074, the statistical power of the correlation is 1 (**Supplementary Fig. 4**). The same method was used to calculate the MRI

features, achieving the statistical power of 1 for fALFF and 1 for GMV respectively, which are all high enough to assure accurate and robust conclusions about the correlations between PRS and MRI loadings detected. We have added the above contents in **Supplementary “Power analysis”** and **“Weakly association between PRS and the identified components”** sections.

t tests – Correlation: Point biserial model			
Analysis:	Post hoc: Compute achieved power		
Input:	Tail(s)	=	One
	Effect size $ \rho $	=	0.074
	α err prob	=	0.05
	Total sample size	=	22773
Output:	Noncentrality parameter δ	=	11.1978387
	Critical t	=	1.6449205
	Df	=	22771
	Power (1- β err prob)	=	1.0000000

Supplementary Figure 4. Statistical power generated from the G*Power software.

6.4 “One explanation may be that the clinical analyses are just picking about global aspects of brain pathology, which could be addressed through the specificity analysis mentioned above.”

The specificity of the identified PRS pattern comparing with the null pattern was also validated. The classification and group difference test based on features from the null pattern (**Supplementary Fig. 4b**) identified from the permuted PRS were validated. As displayed in **Supplementary Fig. 13a**, there are no group differences between SZ and HC of the null pattern, and the classification accuracy is around 50% across the 4 independent SZ cohorts (**Supplementary Fig. 13b**). When comparing with **Fig. 4** (the main text), it is clear that the identified PRS-associated pattern is specific in discriminating between SZ and HC. Please also note that the identified PRS pattern can predict cognition and symptoms for SZ across 3 independent cohorts. The prediction was rigorous, which means that for each clinical measure, the same multiple linear regression model and the same ROIs were used for all the 3 SZ cohorts. Therefore, the identified PRS-associated frontotemporal pattern is robust in predicting cognition (attention, working memory and composite scores) and PANSS negative scores.

(a) PRS associated pattern

(b) Null pattern

Supplementary Figure 4. (a) The fALFF+GMV covarying pattern associated with PRS. (b) The most frequently occurring (voxels with more than 60% occurrences) covarying pattern associated with 500 times permuted PRS.

Supplementary Figure 13. (a) Group differences between SZ and HC of the permuted pattern for independent BSNIP-1, COBRE, fBIRN and MPRC cohorts, respectively. (b) ROC curves of the classification results between SZ and HC for BSNIP-1, COBRE, fBIRN and MPRC cohorts, respectively.

We also added the following statements in discussion.

.....“Despite reliable PRS pattern validation, the PRS was only weakly associated with the components’ loadings, but with high enough statistical power ($1 - \beta = 1$). This is in line with most previous published large sample sized UKB SZ PRS-MRI association studies¹⁵⁻¹⁷”

7. The results section should provide more detail about the methods as when reporting results.

We have added more methods details when reporting related results as follows:

“Schizophrenia PRS were calculated based on Psychiatric Genomics Consortium Schizophrenia (PGC SZ 2) 108 risk loci¹⁹, thresholded at $P_{\text{SNP}} < 5.0\text{e-}08$ and pruned at $r^2 < 0.1$, which followed a normal distribution (**Supplementary Fig. 1**). Head motion, site, gender and age were regressed out from fALFF/GMV feature matrices prior to fusion analysis. Within the healthy Caucasian UKB data (N=22,459, demographic information can be found in **Table 1**), fusion with PRS was performed to identify PRS-associated fALFF+GMV multimodal pattern (details on fusion with reference method can be found in **Methods** section).”

“The robustness of the identified PRS pattern was also validated. The same PRS-guided fusion was performed on different split of UKB sample (healthy Caucasian, healthy subjects and all available subjects that passed MRI quality control) under different P_{SNP} ($5.0\text{e-}08$, $1.0\text{e-}04$, 0.05) and pruning ($r^2 < 0.1$ and 0.2) thresholds.”

“The identified PRS-associated fALFF+GMV components were separated into positive ($Z > 0$) and negative ($Z < 0$) brain networks based on the Z-scored brain maps. Thus 4 PRS-associated brain features (fALFF_positive, fALFF_negative, GMV_positive, GMV_negative) were obtained by averaging fALFF/GMV in these networks. Two sample t-tests were used to estimate the group differences of these 4 PRS features between SZ and HC.”

“The classification ability of the identified PRS-associated brain network was validated by using a linear support vector machine (SVM) approach to classify SZ patients and HCs. In addition to the averaged fALFF/GMV values, the first 5 principal components (PC, obtained from principal component analysis) resulted from decomposing the fALFF/GMV positive/negative feature matrices within the 4 PRS networks were also included as feature input, resulting in 6 features for each (the mean + 5 PC) PRS-associated network, *i.e.*, 24 features in each HC-SZ cohorts (details on feature selection and SVM classification can be found in **Methods** and **Supplementary “Feature selection and classification”** sections). Note that the first 5 PCs captured 99% variance from the identified PRS-associated ROIs, while the mean only represented <50% variance (**Supplementary Fig. 11**).”

“The four mean PRS-associated brain features were further used to construct multiple linear regression models (Eq.2) for each domain from COBRE cohort to predict cognitive and symptom

measures for fBIRN and BSNIP SZ patients (the same models and the same ROIs were used in the generalized prediction). Correlations between the estimated symptom/cognitive scores and its true values were calculated to estimate the prediction performance.”.....

“To test whether the identified PRS-derived pattern was evident in other psychotic disorders, ANOVA and two sample t-tests were used to calculate group differences of the 4 PRS features.....”

“The ability of the identified PRS-associated brain features in differentiating between other neuropsychaitric and mood disorders and HC was also tested by two sample t-tests.”.....

8. I could not find much information about the fusion approach at all.

The fusion details were added in the updated manuscript, see page 27.

“The study design of developing, testing and validating of the PRS-associated multimodal biomarkers was displayed in **Fig. 1**. Firstly, schizophrenia PRS-guided fusion was performed in healthy Caucasian in UKB. Specifically, subject-wise PRS values were used as a reference to jointly decompose the preprocessed fALFF (\mathbf{X}_1) and GMV (\mathbf{X}_2) by “MCCAR+jICA”²⁰⁻²² to investigate PRS-associated multimodal brain network. The correlations of imaging components with PRS was maximized in the supervised fusion method, as in equation (1).

$$\max \sum_{k,j=1}^2 \left\{ \|\text{corr}(\mathbf{A}_k, \mathbf{A}_j)\|_2^2 + 2\lambda \cdot \|\text{corr}(\mathbf{A}_k, \text{PRS})\|_2^2 \right\} \quad (1)$$

where \mathbf{A}_k is the loading matrix for each modality, $\text{corr}(\mathbf{A}_k, \mathbf{A}_j)$ is the column-wise correlation between \mathbf{A}_k and \mathbf{A}_j , and $\text{corr}(\mathbf{A}_k, \text{PRS})$ is the column-wise correlation between \mathbf{A}_k and PRS. This supervised fusion method can extract a joint multimodal component(s) that significantly associated with PRS.”

9. It’s unclear which specific p-threshold for selecting SNPs from the original GWAS was used in the PRS computation. Was the same one used throughout for all analyses?

Thank you for pointing this out. The same GWAS threshold ($p < 5.0e-08$, the strictest threshold) was used to calculate the SZ PRS for UKB, and the 4 independent SZ cohorts. Please also refer to the response for **Reviewer #2 point 8**, we have added different P_{SNP} thresholds (5.0e-08, 1.0e-04, 0.05) to calculate the SZ PRS to show whether the current results are affected by different P_{SNP} thresholds.

10. There are grammatical errors throughout. Please check. E.g., Line 63 – should be studies; Line 66 – should be ‘represents’ ... and several others elsewhere.

Thank you for pointing this out. We have gone through the whole manuscript and corrected the

grammatical errors.

Reviewer #2 (Remarks to the Author):

Summary: NCOMMS-21-43466 describes a large cross-sectional association study between polygenic risk score (PRS) for schizophrenia (SZ) and multi-modal MRI (T1w & resting state fMRI). Authors establish robust associations between PRS and both modes of MRI data in a large, well-powered sample ($N > 22,000$). The authors further use these MRI profiles as brain features to detect / predict SZ case-control differences in several independent cohorts. Their results suggest that a pattern of SZ-PRS related structure-function alterations are generalizable, performing well as a classifier in several naïve datasets as well associating with cognition and SZ symptom profiles. The authors further show relative disease specificity – predominately linking the feature profile to psychotic disorders, but not mood / neurodevelopment disorders.

Thank you for the careful reading, helpful comments, and constructive suggestions, which have significantly improved the presentation of the manuscript.

1. Patterns of associations between SZ PRS and MRI data in UK Biobank have been reported before for both T1w (PMID: 34462574) and ALFF modes (PMID: 32514083), making this part of the investigation largely confirmatory / repeated. The merit / novelty of the current investigation is in demonstrating how these PRS – MRI patterns are observable SZ cases in other datasets. While this is also not the first study to show that (discovery) SZ -related brain effects are present in independent (test) samples (e.g PMID: 32463560), it is (to my knowledge), the first to show that PRS-brain associations can distinguish SZ case/control differences – which may inform more about risk that SZ associated confounding. There are a number of points that need further exploration and additional clarification.

Thank you for the constructive comments. The recommended references were added to the introduction. We have also updated the introduction and discussion according to the above comments.

.....“Although brain abnormalities linked with PRS have been reported in schizophrenia, prior studies focus on a single imaging modality and used a region of interest (ROI) based simple correlation analysis. There have, to the best of our knowledge, been no fusing of whole brain MRI studies to identify PRS-associated multimodal brain abnormalities, including the use of machine learning methods to assess its biomarker properties^{23,24}. More specifically, there have been neither joint PRS-multimodal brain imaging studies focused on the classification of SZ and healthy controls (HC), nor use of these variables to predict cognition or symptoms.”.....

.....“To the best of our knowledge, this is the first study to establish and evaluate PRS-associated multimodal neuroimaging biomarkers with rigorous cross-site classification and prediction, which may inform more about genetic risk that SZ associated confounding.”

.....“Furthermore, the extant literature in the field of PRS and neuroimaging association studies in SZ has mainly focused on pathophysiology explanations of SZ, while our current study goes beyond PRS and neuroimaging associations by testing its ability in classification and prediction, as well as demonstrate its diagnostic heterogeneity within psychosis (frontotemporal alterations were most severe in schizophrenia and schizo-affective, milder in bipolar), and indistinguishable from HC in autism, depression and attention-deficit hyperactivity disorder. This study helps moving forward from single modality, ROI based associations with PRS to multimodal and the whole brain data-driven analysis by both cross cohorts’ prediction and classification validation. All above findings indicate the potential of the identified PRS-associated multimodal frontotemporal network to serve as a transdiagnostic gene intermediated brain signature specific to SZ, which underlies the genetic pathophysiology of schizophrenia and also provide the first evidence for its potential to be used as a biomarker.”

2. Line 60 - Reference 1 does not relate to epidemiology observations of SZ prevalence, but to a prior UKBB PRS-SZ brain imaging study.

Thank you for pointing this out. Reference 1 was removed from the first sentence in introduction.

3. Line 64 – Accounts of SZ PRS should be to explain overall liability (which I believe is currently a little over 7%, PMID: 25056061), rather than diagnostic variance).

Thank you for the valuable comments. We have updated line 64 accordingly and cite the recommend reference PMID: 25056061:

16. Ripke, S., *et al.* Biological insights from 108 schizophrenia-associated genetic loci. *Nature* **511**, 421-+ (2014).

.....“Genome wide association studies discovered many SZ-related risk loci that account for ~25% of the diagnostic variance have now been identified^{25,26}, although the effect size of any single locus is small (~7%)¹⁹.”.....

4. Line 79 – Polygenic risk for SZ has been associated with brain alterations the cite references [21-24], none of which support a relationship between MRI IPDs and SZ-PRS, especially Franke et al., 2016 [23].

We have corrected for the right reference: T1w (PMID: 34462574) and ALFF modes (PMID: 32514083), PMID: 32463560 as the reviewer recommended before.

5. Lines 174-185 – I do not think the term ‘replicable’ is applicable here, as the authors are making the inference on an extended pool of participants, using different parameters. I would suggest the use of a split / independent sample would be more appropriate to make this inference. I would suggest this would be useful either way, to ensure the fusion model / pattern is not overfitting the data.

We have changed the term “replicable” to “the use of a split sample”.

6. Line 181 – well replicated (>80%) with different PRS thresholds is misleading – the only thresholds used are various pruning strategies, not P-thresholds (supplementary material). See also query for 405-409.

Thank you for the valuable comments. We have added different GWAS P_{SNP} thresholds to calculate the PRS and to check whether the identified PRS-associated pattern can be replicated. Please refer the details in the response for **Reviewer #2 point 8**.

7. Line 224 (and throughout manuscript) – There are well-established negative associations between SZ - PRS and intelligence / cognition (e.g. PMID: 34347035). While The PRS-brain features maybe specific to SZ, how much of this variance can be explained by reduced IQ in UKBB participants with high SZ-PRS and lower IQ in the patient samples?

Thank you for the comments.

PMID: 34347035. Legge, S.E., *et al.* Associations Between Schizophrenia Polygenic Liability, Symptom Dimensions, and Cognitive Ability in Schizophrenia. *JAMA psychiatry* **78**, 1143-1151 (2021).

The paper recommended above found that schizophrenia PRS and intelligence PRS were correlated with cognitive ability. The **intelligence PRS was strongly associated with IQ**, whereas the schizophrenia PRS was not correlated with IQ²⁷. So, the well-established associations identified in PMID: 34347035, were between SZ-PRS and cognition, not SZ-PRS and IQ (intelligence).

Here, in response to the reviewer, we calculated the correlation between SZ-PRS and IQ in UKB cohort (for SZ patient cohorts, IQ scores were not available). Results (**Supplementary Fig. 4**) showed that there were moderate association between SZ-PRS and IQ within UKB. So the variance explained by reduced IQ in UKB participants (R^2) was 0.09%~0.12%.

Supplementary Figure 4. 2D density plot of the correlation between SZ PRS and IQ for both PGC SZ 2 (108 loci) and PGC SZ 3 (270 loci).

8. Line 405 - 409 – It was not clear why the authors chose to restrict their SZ PRS to SNPs that surpassed GWAS threshold ($P < 5 * 10^{-8}$) when more liberal P-threshold (e.g $P < 0.05$) explain more variance in case-control status (PMID: 25056061).

Sorry for the lack of clarity. Yes, we did use $P_{\text{SNP}} < 5.0e-08$ as the threshold to select SNPs to calculate SZ PRS in our current study, which was the strictest threshold usually applied in SZ GWAS analysis²⁸. Here, in response to the reviewer, we also include the fusion with PRS results based on different thresholds: $P_{\text{SNP}} < 5.0e-08$, $P_{\text{SNP}} < 1.0e-04$ and $P_{\text{SNP}} < 0.05$. Briefly, three different P_{SNP} ($5.0e-08$, $1.0e-04$, 0.05) were used to generate SZ PRS, which were used as references to supervise fALFF and GMV fusion. There was high spatial overlap ($>50\%$) among these PRS-associated patterns between $P_{\text{SNP}} < 5.0e-08$ and $P_{\text{SNP}} < 1.0e-04$, $P_{\text{SNP}} < 0.05$ (details can be found in the **Supplementary “Spatial similarity”** section).

Supplementary Figure 5. Fusion with PRS under different P_{SNP} thresholds. (a) $P_{\text{SNP}} < 5.0e-08$; (b) $P_{\text{SNP}} < 1.0e-04$; (c) $P_{\text{SNP}} < 0.05$. The spatial similarity among these PRS-associated patterns are displayed in the following table.

Spatial similarity between Fig. S5a and Fig. S5b-c.

Correlation (r value)	Fig. S5b	Fig. S5c
Fig. S5a fALFF	0.53	0.64
Fig. S5a GMV	0.72	0.58

The positive fALFF in MIFC, SMTc, negative fALFF in PCC and MOC, accompanied with positive GMV in anterior insula and hippocampus, and negative GMV in MITC, and parahippocampus were all well replicated (>50%) under different P_{SNP} thresholds.

The above contents were added into Supplementary file. The study design figure (**Fig. 1**) was also updated accordingly (page 5).

9. Lines 412-414 – I am slightly concerned about the FD threshold used for exclusion criteria of head motion. A mean FD of 2.5mm (nearly 1 voxel) is extremely liberal censoring protocol (typically > 0.2mm mean FD for rsfMRI, PMID: 29440410)

Thank you for the comments. We have added the following experiments with regard to the effect of head motion.

9.1 Motion on preprocessing

Please note that in the preprocessing procedure for fMRI, we despiked the fMRI data: nuisance covariates (6 head motions + cerebrospinal fluid [CSF] + white matter [WM]) + global signal were regressed out via a general linear model from the voxel time series. We did remove outlier subjects who have framewise displacements (FD) exceeding 1.0 mm, as well as head motion exceeding 2.5 mm of maximal translation (in any direction of x, y or z) or 1.0° of maximal rotation throughout the course of scanning. Results indicate all FDs (mean framewise displacements, mean of root of mean square frame-to-frame head motions assuming 50 mm head radius⁵) for all subjects were 1 mm at every time point. In addition to denoising the BOLD fMRI signal in preprocessing, ICA approach used in current study can further isolate artifact components (motion and noise) from fMRI⁶.

2.5 mm is the maximal translation in any direction of x, y or z, not the mean FD. The mean FD for all subjects are $<1\text{mm}$.

9.2 PRS pattern on UKB subset with head motion $<0.2\text{mm}$

As requested from the reviewer, we also exclude subjects with $>0.2\text{mm}$ FD to get a subset of UKB (N = 13490, 60% subjects' head motion $<0.2\text{mm}$) to perform the fusion with PRS to test whether the identified multimodal frontotemporal pattern can be replicated. Result (**Supplementary Fig. 10**) showed that the identified PRS-associated pattern (frontotemporal cortex and thalamus in fALFF, accompanied with thalamus, hippocampus, para-hippocampus and temporal cortex in GMV) can be validated on UKB subset with mean FD $<0.2\text{mm}$. This means that head motion is not a major confounding factor for the current fusion results.

Supplementary Figure 10. (a) The original PRS-associated fALFF+GMV covarying pattern for UKB. (b) The PRS pattern on UKB subset with head motion $<0.2\text{mm}$.

9.3 Group differences of mean FD between SZ and HC

We have calculated the group differences of mean FD between HC and SZ across the 4 SZ cohorts included in this study. Note that there were no significant differences between patients and controls on mean FD for all the 4 SZ cohorts, namely,

BSNIP, HC: mean=0.22±0.11mm, SZ: 0.26±0.24mm, two sample t-test: $p = 0.25$

FBIRN, HC: mean=0.25±0.18mm, SZ: 0.27±0.21mm, two sample t-test: $p = 0.65$

COBRE, HC: mean=0.22±0.12mm, SZ: 0.21±0.11 mm, two sample t-test: $p = 0.77$

MPRC, HC: mean=0.12±0.22mm, SZ: 0.18±0.10mm, two sample t-test: $p = 0.15$

9.4 Partial correlation

Partial correlation has been proposed as an alternative approach for removing spurious shared variance in correlation analysis⁷. Here, we also performed partial correlation analysis between the identified component and PRS by regressing out mean FD. Result showed that the significant level was not changed by the mean FD ($p = 5.2e-30^*$ for fALFF, $p = 2.3e-28^*$ for GMV as in **Fig. 2b**).

9.5 fALFF not functional connectivity

PMID: 29440410. Power, J.D., *et al.* Ridding fMRI data of motion-related influences: Removal of signals with distinct spatial and physical bases in multiecho data. *Proceedings of the National Academy of Sciences of the United States of America* **115**, E2105-E2114 (2018).

The above recommended paper found that head motion was sensitive to functional connectivity analysis. However, the current fusion analysis was conducted on the spatial maps of fALFF not functional connectivity. While it is the functional connectivity derived from rs-fMRI that is more sensitive to head motion⁸⁻¹².

Collectively, considering there was no group difference in head motion between SZ and HC, and no significant correlation between mean FD and PRS, and the partial correlation between the identified component and PRS remained significant after regressing out mean FD, and the PRS-pattern can be replicated on UKB subset with head motion <0.2mm, the current fusion analysis was based on fALFF not functional connectivity, we believe that micro-motion is not a major factor affecting the current results.

The motion related paper referred by the reviewer were cited in the revised manuscript, and the above contents were added in **Supplementary “Motion effect”** section and the main text **“Site and motion effects on the identified PRS-pattern”** section (page 12-13), respectively.

10. Lines 416 - mean FD ~ PRS correlation $P > 0.5$, do authors mean $P > 0.05$, as association between Mean FD and PRS0.1 was $p = 0.09$ (supplementary, line 245)?

Thank you for the carefully comments. Line 416 was updated as following:

“The correlation between mean FD and SZ PRS is not significant ($p > 0.05$, Supplementary Table 5).”

References

1. Insel, T., *et al.* Research Domain Criteria (RDoC): Toward a New Classification Framework for Research on Mental Disorders. *American Journal of Psychiatry* **167**, 748–751 (2010).
2. Santelmann, H., Franklin, J., Busshoff, J. & Baethge, C. Interrater reliability of schizoaffective disorder compared with schizophrenia, bipolar disorder, and unipolar depression – A systematic review and meta-analysis. *Schizophrenia research* **176**, 357–363 (2016).
3. McGorry, P.D., *et al.* Spurious Precision – Procedural Validity of Diagnostic-Assessment in Psychotic Disorders. *American Journal of Psychiatry* **152**, 220–223 (1995).
4. Radua, J., *et al.* Increased power by harmonizing structural MRI site differences with the ComBat batch adjustment method in ENIGMA. *Neuroimage* **218**, 116956 (2020).
5. Allen, E.A., *et al.* A baseline for the multivariate comparison of resting-state networks. *Frontiers in systems neuroscience* **5**, 2 (2011).
6. Du, Y., *et al.* Artifact removal in the context of group ICA: A comparison of single-subject and group approaches. *Human brain mapping* **37**, 1005–1025 (2016).
7. Smith, S.M., *et al.* Network modelling methods for FMRI. *Neuroimage* **54**, 875–891 (2011).
8. Satterthwaite, T.D., *et al.* An improved framework for confound regression and filtering for control of motion artifact in the preprocessing of resting-state functional connectivity data. *Neuroimage* **64**, 240–256 (2013).
9. Satterthwaite, T.D., *et al.* Impact of in-scanner head motion on multiple measures of functional connectivity: relevance for studies of neurodevelopment in youth. *Neuroimage* **60**, 623–632 (2012).
10. Ciric, R., *et al.* Benchmarking of participant-level confound regression strategies for the control of motion artifact in studies of functional connectivity. *Neuroimage* **154**, 174–187 (2017).
11. Ciric, R., *et al.* Mitigating head motion artifact in functional connectivity MRI. *Nature protocols* **13**, 2801–2826 (2018).
12. Power, J.D., *et al.* Ridding fMRI data of motion-related influences: Removal of signals with distinct spatial and physical bases in multiecho data. *Proceedings of the National Academy of Sciences of the United States of America* **115**, E2105–E2114 (2018).
13. Zou, Q.H., *et al.* An improved approach to detection of amplitude of low-frequency fluctuation (ALFF) for resting-state fMRI: fractional ALFF. *Journal of neuroscience methods* **172**, 137–141 (2008).
14. Turner, J.A., *et al.* A multi-site resting state fMRI study on the amplitude of low frequency fluctuations in schizophrenia. *Front Neurosci* **7**, 137 (2013).
15. Neilson, E., *et al.* Impact of Polygenic Risk for Schizophrenia on Cortical Structure in UK Biobank. *Biological psychiatry* **86**, 536–544 (2019).

16. Stauffer, E.M., *et al.* Grey and white matter microstructure is associated with polygenic risk for schizophrenia. *Molecular psychiatry* (2021).
17. Marek, S., *et al.* Reproducible brain-wide association studies require thousands of individuals. *Nature* **603**, 654–660 (2022).
18. Faul, F., Erdfelder, E., Lang, A.G. & Buchner, A. G*Power 3: a flexible statistical power analysis program for the social, behavioral, and biomedical sciences. *Behavior research methods* **39**, 175–191 (2007).
19. Ripke, S., *et al.* Biological insights from 108 schizophrenia-associated genetic loci. *Nature* **511**, 421–+ (2014).
20. Qi, S., *et al.* Reward Processing in Novelty Seekers: A Transdiagnostic Psychiatric Imaging Biomarker. *Biological psychiatry* **90**, 529–539 (2021).
21. Qi, S., *et al.* Multimodal Fusion With Reference: Searching for Joint Neuromarkers of Working Memory Deficits in Schizophrenia. *IEEE Trans Med Imaging* **37**, 93–105 (2018).
22. Qi, S., *et al.* MicroRNA132 associated multimodal neuroimaging patterns in unmedicated major depressive disorder. *Brain* **141**, 916–926 (2018).
23. Abi-Dargham, A. & Horga, G. The search for imaging biomarkers in psychiatric disorders. *Nat Med* **22**, 1248–1255 (2016).
24. Li, A., *et al.* A neuroimaging biomarker for striatal dysfunction in schizophrenia. *Nature medicine* **26**, 558–565 (2020).
25. Pardinas, A.F., *et al.* Common schizophrenia alleles are enriched in mutation-intolerant genes and in regions under strong background selection. *Nat Genet* **50**, 381–389 (2018).
26. Riglin, L., *et al.* Schizophrenia risk alleles and neurodevelopmental outcomes in childhood: a population-based cohort study. *Lancet Psychiatry* **4**, 57–62 (2017).
27. Legge, S.E., *et al.* Associations Between Schizophrenia Polygenic Liability, Symptom Dimensions, and Cognitive Ability in Schizophrenia. *JAMA psychiatry* **78**, 1143–1151 (2021).
28. Consortium, S.W.G.o.t.P.G., Ripke, S., Walters, J.T.R. & O’ Donovan, M.C. Mapping genomic loci prioritises genes and implicates synaptic biology in schizophrenia. *Preprint at medRxiv*. <https://doi.org/10.1101/2020.09.12.20192922>. (2020).

Reviewers' comments:

Reviewer #1 (Remarks to the Author):

The authors have addressed all comments thoroughly.

Reviewer #2 (Remarks to the Author):

The authors have provided a substantially improved manuscript, addressing many methodological concerns addressed by myself and reviewer 1 (including confounding from site, motion and PRS-thresholding effects) to demonstrate consistent associations for the inferences initially reported.

I have two remaining issues that require clarification.

R2.5 - I recommended that if the authors wanted to describe their observations as 'replicable' they would need to use a split-sample (i.e. split their sample and test associations independently), rather than re-label their analysis as a 'split-sample'. However, the new site-specific analysis provided (supplementary figure 9), does provide this evidence. I suggest re-wording to "PRS-pattern consistency across PRS parameters within UKB sample"

R2.7 - I suggested the author control for SZ-PRS ~ IQ associations in the MRI data analysis. The authors reproduce robust SZ-PRS ~ IQ associations in UKBB, but do not control / consider this in their MRI-analysis. It would be useful to measure how much variance in the SZ-PRS MRI profiles is explained by IQ and how the accuracy of the classifier is changed after this consideration.

REVIEWERS' COMMENTS:

Reviewer #1 (Remarks to the Author):

The authors have addressed all comments thoroughly.

Thank you for the comments.

Reviewer #2 (Remarks to the Author):

The authors have provided a substantially improved manuscript, addressing many methodological concerns addressed by myself and reviewer 1 (including confounding from site, motion and PRS-thresholding effects) to demonstrate consistent associations for the inferences initially reported.

Thank you for the supportive comments.

I have two remaining issues that require clarification.

R2.5 - I recommended that if the authors wanted to describe their observations as 'replicable' they would need to use a split-sample (i.e. split their sample and test associations independently), rather than re-label their analysis as a 'split-sample'. However, the new site-specific analysis provided (supplementary figure 9), does provide this evidence. I suggest re-wording to "PRS-pattern consistency across PRS parameters within UKB sample".

Thank you for the suggestions. The subtitle of “PRS-pattern validation with different split of UKB sample” was revised as “PRS-pattern consistency across PRS parameters within UKB sample” as suggested by the reviewer.

R2.7 - I suggested the author control for SZ-PRS ~ IQ associations in the MRI data analysis. The authors reproduce robust SZ-PRS ~ IQ associations in UKBB, but do not control / consider this in their MRI-analysis. It would be useful to measure how much variance in the SZ-PRS MRI profiles is explained by IQ and how the accuracy of the classifier is changed after this consideration.

Thank you for the comments. Note that the significance level of the correlation between PRS-SZ and IQ ($p \sim e-05$, not FDR corrected, **Supplementary Fig. 4**) is relatively low when comparing with the PRS-MRI associations ($p \sim e-28^*$, **Fig. 2b**) as in our results. It is the **intelligence PRS was strongly associated with IQ**, whereas the schizophrenia PRS was not correlated with IQ¹. This is consistent with our current big data analysis. The well-established associations identified in PMID: 34347035, were between SZ-PRS and cognition, not SZ-PRS and IQ (intelligence).

Furthermore, we also calculated the partial correlations between loadings and PRS after

regressing out IQ. Results showed the significance level of the correlations between loadings and PRS did not affected by IQ ($p=4.2e-30^*$ and $p=1.7e-28^*$ for fMRI and sMRI respectively).

Note that age, gender, mean FD and site were regressed out from fALFF+GMV matrices prior to PRS-guided fusion. Here, as requested from the reviewer, we also performed the PRS-guided fusion analysis after regressing out age, gender, mean FD, site, as well as IQ. Results showed that the original PRS pattern (**Supplementary Fig. 10a**) and the IQ regressed out pattern (**Supplementary Fig. 10b**) are highly similar (the spatial overlapping between them is >0.9 for both fALFF and GMV components). Thus the accuracy of classifier would not be affected after regressing out IQ, since the patterns are almost the same.

Considering that SZ-PRS is marginally correlated with IQ ($p\sim e-05$, not FDR corrected), the significance level of the correlations between loadings of the identified component and PRS did not affected by IQ ($p=4.2e-30^*$ and $p=1.7e-28^*$ for fMRI and sMRI respectively), the PRS pattern can be replicated after regressing out IQ (**Supplementary Fig. 10**), we believe that IQ is not a major factor affecting the current results.

Supplementary Figure 10. (a) The original PRS-associated fALFF+GMV covarying pattern for UKB. (b) The PRS pattern on UKB subset with head motion $< 0.2\text{mm}$.

Supplementary Figure 4. 2D density plot of the correlation between SZ PRS and IQ for both PGC SZ 2 (108 loci) and PGC SZ 3 (270 loci).

Figure 2. The identified PRS-associated multimodal joint components in UKB healthy whites (N=22,459). (a) Spatial brain maps visualized at $|Z| > 2$. (b) 2D density plot of PRS and loadings of components in UKB ($p=4.1e-30^*$ and $p=1.6e-28^*$ for fMRI and sMRI respectively). (c) Correlations ($p=1.2e-04^*$ and $p=1.4e-04^*$ for fMRI and sMRI respectively) between projected (projecting spatial maps from UKB to SZ patients to obtain an estimation of their mixing matrices) loadings and PRS within SZ patients (N=290, where PRS was available). Pearson correlation analysis was used to calculate the correlation between PRS and loadings in a-b. Source data are provided as a Source Data file.

1. Legge, S.E., *et al.* Associations Between Schizophrenia Polygenic Liability, Symptom Dimensions, and Cognitive Ability in Schizophrenia. *JAMA psychiatry* **78**, 1143–1151 (2021).